# The Effects of Changing Climate on Estuarine Water Levels: A United States Pacific Northwest Case Study

Kai Parker[1], David Hill[2], Gabriel García-Medina[3], Jordan Beamer[4]

[1]School of Civil and Construction Engineering, Oregon State University, Corvallis Oregon, 97330, USA
[2] School of Civil and Construction Engineering, Oregon State University, Corvallis Oregon, 97330, USA
[3] Marine Sciences Laboratory, Pacific Northwest National Laboratory, Seattle Washington, 98109, USA
[4] Oregon Water Resources Department, Salem Oregon, 97301, USA

*Correspondence to*: Kai Parker (parkerk@oregonstate.edu)

**Abstract.** Climate change impacts to extreme water levels (WLs) at two United States Pacific Northwest estuaries are investigated using a multi-component process-based modeling framework. The integrated impact of climate change on estuarine forcing is considered using a series of sub-models that track changes to oceanic, atmospheric, and hydrologic controls on hydrodynamics. This modeling framework is run at decadal scales for historic (1979-1999) and future (2041-2070) periods with changes to extreme WLs quantified across the two study sites. It is found that there is spatial variability in extreme WLs at both study sites with all recurrence interval events increasing with further distance into the estuary. This spatial variability is found to increase for the 100-year event moving into the future. It is found that the full effect of sea level rise is mitigated by a decrease in forcing. Short recurrence interval events are less buffered and therefore more impacted by sea level rise than higher return interval events. Finally, results show that annual extremes at the study sites are defined by compound events with a variety of forcing contributing to high WLs.

## 1. Introduction

Estuaries are important intersections of human and natural systems, serving as both some of the most resource-rich ecosystems on Earth and some of the most densely populated. Asked to meet many, at times conflicting, needs, estuaries require careful management. Unfortunately, coastal planning is limited by an insufficient understanding of how estuaries will respond to future conditions. In particular, extreme water levels (WLs) are of first-order importance with flooding putting both lives and physical infrastructure at risk.

In the United States (U.S.) Pacific Northwest (PNW), considerable progress has been made towards an understanding of hazard risk at open beaches as controlled by a combination of forcing drivers, including waves, tides, winds, and others [Barnard et al., 2014; Ruggiero, 2013; Serafin and Ruggiero, 2014]. Flooding risk in PNW estuaries is less well understood, primarily due to the greater complexity of the estuarine environment. Research efforts have mostly focused on the Colombia river, which is

societally important, but not necessarily characteristic of other PNW estuaries due to its large size and heavy damming/flow regulation [Jay et al., 2015,2016; Lee et al., 2009]]. Estuarine hydrodynamics remain more complicated than open coastlines due to the additional driver of streamflow and a much more complicated topographical context (e.g. embayment and complex bathymetry) [Odigie and Warrick, 2018; Wahl et al., 2015]. This makes estuaries difficult to simplify as they exhibit nonlinear water column response [Ding et al., 2013] with forcing contributions being difficult to uncouple [Wolf, 2009].

Consideration of future risk adds an additional challenge through a mismatch in time scales. Climate change signals are significant at scales of decades while individual extreme events occur at timescales of hours to days. This requires both high model temporal resolution and long model simulations. Long time series are additionally necessary for extreme value analysis and constraining event recurrence intervals (RIs). Balancing these needs with computational cost has remained a major obstacle and has led to a variety of modeling strategies. One approach takes advantage of the fact that extreme value analysis only requires information of the maxima events. Lin et al. [2012] use a unique approach based on multiple model complexities; an efficient model to determine which events will cause flooding and a more complex model to accurately quantify the dynamics of those events. As a second example, Orton et al. [2016] simply model the coastal responses to a large set of hurricane events. Both of these studies focus on the eastern U.S. coastline where tropical cyclones are the principal driver of flooding events. Locations less dominated by tropical cyclones have a more diverse and balanced set of contributions to flooding [Parker, 2019]. In these locations, extreme events can occur due to combinations of forcings which are not individually extreme, a phenomenon discussed in the literature as "compound events" [Gallien, et al., 2018; Leonard et al., 2014; Moftakhari et al., 2017; Wahl et al., 2015]. This makes it difficult to know a priori which events will result in maximum WLs.

Recent advances in computing power and parallel processing have opened up an alternative possibility of running continuous time series hydrodynamic models at climate change scales (decades to centuries). This allows examination of extremes without assuming which events will cause extremes. Additionally, a continuous time series analysis has other desirable properties. For example, an event based approach limits analysis to only large RI events, eliminating information on higher probability, lower magnitude, events. This is undesirable since so-called "nuisance flooding" can, over time, lead to a higher aggregate cost than extreme events, especially when considering sea level rise (SLR) [Maftakhari et al., 2015; 2017]. Additionally, a continuous time series approach also allows an integrated consideration of the coupled effect of the various changing controls on estuarine hydrodynamics. Many studies have focused on individual components of climate change (e.g., just SLR) but few have addressed their combined effects on estuarine flooding. This is problematic since there can be interactions among processes. As one example, SLR has been shown to non-linearly modulate storm surge [Smith et al., 2010].

Studies that have attempted to holistically model estuarine flooding along the U.S. West coast include Cloern et al. [2011] who studied century-scale change in San Francisco Bay. Their study used a hybrid approach, coupling process-based and statistical sub-models to evolve water column properties over time. Barnard et al.'s [2014] Coastal Storm Modeling System (CoSMoS) also used an interlinked model framework but with less focus on estuary WLs (although CoSMoS 2.0 improves upon this). Cheng et al. [2015a] did a preliminary study of a single PNW estuary using a fully process-based model framework. The current study builds upon this effort by including additional physical processes, conducting a comparative study, and applying the results to the production of flood mapping products.

The objective of the present study is to further develop this research direction through applying a comprehensive process-based modeling framework to the problem of estuarine flooding under current and future conditions. A process-based approach allows direct modeling of climate induced changes to all drivers (streamflow, wave forcing, etc.) of estuarine WLs. This study hypothesizes that considering integrated forcing on estuaries results in significant spatial variability in extreme WLs. This hypothesis is in contrast to the static "bath-tub" approximation (i.e., the assumption of a horizontal water surface), which is commonly used despite having been shown to potentially result in significant errors [Gallien et al., 2014]. Results from this study quantify this error as well as provide information on how it may be evolving as a result of climate change. Additionally, flood surface information will be combined with a high resolution digital elevation model (DEM) to determine the extent of flooding and how it may be changing over time.

This paper is organized with the initial section providing a description of the two study sites (Section 2). The overall modeling framework is then introduced (Section 3) with description of the individual sub-model components. This is followed by information on nonstationary RIs (Section 4). The paper then moves on to the results (Section 5) and closes with a discussion of findings (Section 6).

## 2. Study Sites

This study focuses on two U.S. PNW estuaries, Coos and Tillamook Bays (Figure 1). These estuaries were selected for two reasons. First, each has an active watershed / estuary organization (Coos Watershed Association / Tillamook Estuaries Partnership) which allowed for data sharing and project collaboration. Secondly, the estuaries have similar forcing profiles (Table 1) which maintains some degree of comparability between the two locations. However, the estuaries have very different physical characteristics which allows some exploration into the importance of local bay configuration.

In terms of physical layout, Coos has a unique hook shape while Tillamook has a more classical bay form with an enclosing sand spit defining the western edge (Figure 1). Both estuaries have a jettied inlet and a channel maintained by the U.S. Army Corps of Engineers. Tillamook's entrance is maintained at 5.5 meters deep and 60 meters wide while Coos has a significantly larger deep-draft channel at 18 meters deep and 210 meters wide. Coos has a larger average depth and surface area, resulting in a

greater estuary volume than Tillamook by around 130 million cubic meters. Coos is also narrower with a deep channel along the majority of its length. Tillamook, on the other hand, has a channel only near the entrance with the bulk of its area being defined by broad shallow tidal flats.

In terms of forcing, the estuaries are quite similar although with Tillamook experiencing slightly higher environmental forcing. Tillamook's tidal range is approximately 20 cm larger than Coos and mean wave, wind, and streamflow forcing are all modestly more intensive.

## 3. Methods

This study utilizes a suite of models and data sources to determine the hydrodynamic response of the
study sites to climatic forcing. The overall workflow is that an Atmosphere-Ocean Global Climate Model (AOGCM) serves as the "parent" model providing forcing to a suite of "child" models. These in turn provide the forcing to a hydrodynamic model focused on the estuaries themselves. This modeling framework is conceptually illustrated in Figure 2. In terms of final-stage output, this modeling chain produces continuous time series of WLs (and other variables) at chosen output points. Provided that a
sufficiently long-term simulation is carried out, this spatially explicit information allows for the development of flooding inundation maps at a variety of RIs. The following sections describe each component of the model framework identified in Figure 2.

As is common in climate change impact studies, this study uses paired simulations with hindcast and
forecast boundary conditions. Two simulations were carried out; one for the period 1979-1999 (historic) and the other for the period 2041-2070 (future). The historic period is forced with model output rather than direct observations to control for biases in the AOGCM and modeling framework.

### 3.1 Climate Model

This study uses model data from the North American Regional Climate Change Assessment Program
(NARCCAP) [Mearns et al., 2009, 2013]. NARCCAP provides an ensemble of AOGCMs paired with higher resolution Regional Climate Models (RCMs) focused on the North American continent. The project's future runs are forced by the Special Report on Emissions Scenarios (SRES) A2 emissions scenario which represents one of the higher emissions, anthropologically controlled climate scenarios for the fourth IPCC assessments report [Nakicenovic et al., 2000]. This scenario was chosen (by
NARCCAP) as a conservative but plausible climate trajectory that is in-line with current emissions / population patterns. There are other downscaled climate products available (e.g., MACA; Abatzoglou [2013]) that are based on more current IPCC 5th assessment scenarios; however, NARCCAP was the only climate product, at the start of this project, that provided the necessary offshore coverage with the higher resolution RCM. Most other products were masked (and still are) so that data were only available
on land surfaces while this project required information across the ocean as well. Spatial resolution for

models within NARCCAP is 50 km for RCM variables and range from 1-4 degrees latitude-longitude for the AOGCM models.

While the usage of NARCCAP data forces this project's reliance on an older climate scenario, this does not mean that results are out of alignment with current climate projections. Rather, the A2 SRES scenario is well within the variability of the new scenarios framework of the IPCC 5th assessment. A direct comparison of IPCC 4th and 5th assessment climate scenarios is impossible due to a conceptual change in how scenarios are handled [Nakicenovic et al., 2014; O'Neill et al., 2013]. However, work by Van Vuuren and Carter [2014] has shown that the A2 SRES scenario approximately maps to the representative concentration pathway (RCP) 8,5 and shared socio-economic pathway (SSP) 3 scenario. Since the publication of the IPCC 4th assessment, baseline emissions have been within the range presented within the SRES scenarios [IPCC, 2007] with emissions tracking closer to the higher range of scenarios [The Copenhagen Diagnosis, 2009]. This supports the usage of the A2 scenario for near term projections.

From the model pairings available in NARCCAP, the Community Climate Science Model/Canadian Regional Climate Model (CCSM/CRCM) combination was chosen as providing the best agreement with local in-situ meteorological data. The AOGCM component of this pairing was used for datasets requiring larger spatial coverage while the more finely resolved (both temporally and spatially) RCM was used for nearshore or terrestrial sub-models. When using RCM data, it was found that atmospheric parameters were biased so a univariate statistical bias correction (quantile mapping, [Déqué, 2007]) was performed. The "target" dataset used for the bias correction was the North American Regional Reanalysis (NARR) [Mesinger et al., 2006] dataset interpolated to the location of the CRCM grid nodes. AOGCM data was also found to be biased but was not bias corrected as there is no target dataset that spans the full global climate model extent. Instead this bias correction was handled within the relevant sub-model.

### 3.2 Wave Model

A basin-scale Wavewatch III v3.14 (WW3) [Tolman, 2009] simulation was performed in order to characterize wave climate and provide offshore wave boundary conditions for the two hydrodynamic model domains. WW3 has seen significant success in the U.S. PNW for reproducing wave conditions [e.g., Garcia-Medina et al., 2013] and is well suited to application at large scales [Hanson et al., 2009]. The model was run with two nested grids based on the National Centers for Environmental Prediction (NCEP)'s operational Global and North East Pacific models with a resolution of (1° by 1.25°) and (0.25° by 0.25°) respectively. The model was configured with default options and the Tolman and Chalikov [1996] source term package.

Wind forcing for the WW3 model was provided by the CCSM global model. Wave model predictions were found to exhibit a significant bias in comparison to observed wave parameters in the study areas. This bias is likely a result of the low-resolution wind fields [Holthuijsen et al., 1996] or the AOGCM's

inability to reproduce marine winds [Hemer et al., 2011]. The transfer of bias from wind fields to wave model output has been similarly reproduced and discussed in other studies [Feng et al., 2006, Hemer et al., 2011]. Sensitivity analysis showed that running the hydrodynamic model with over-predicted wave heights produced unrealistic flooding values and overwhelmed the influence of other signals contributing to extreme WLs. Therefore, a bivariate statistical bias correction technique [Piani and Haerter, 2012] was used that corrects both the marginal distribution of wave height and period as well as maintains the correlation structure [Parker and Hill, 2017].

### 3.3 Hydrological Model

Streamflow inputs were developed using a series of weather, snowmelt, and hydrological routing models. Specifically, the MicroMet / SnowModel / HydroFlow suite of spatially distributed models were used [Liston & Elder, 2006a; Liston and Elder, 2006b; Liston and Mernild, 2012]. Readers are directed to the source citations for full details of the models. In summary, this suite distributes relevant meteorological forcing variables, computes the surface energy-balance to simulate snowpack evolution and melt, and then uses a simple linear reservoir-routing procedure to route the runoff from rainfall and snowmelt across the landscape to the coastline. This modeling suite allows for both high temporal and spatial resolution of hydrologic processes (for this study, 100-m grid cell size and a 3-hourly time step). Cheng et al. [2015a] successfully applied this modeling suite to the PNW and Beamer et al. [2017] to the U.S. Gulf of Alaska watershed under climate change scenarios.

The models were calibrated and validated using the NARR dataset for meteorological forcing. Only the Tillamook Bay watershed contains active stream gauges. Therefore, the model was calibrated to observed streamflow at the available gauge locations, the Wilson, Trask and Miami Rivers. After calibration, validation against observed streamflow yielded high Nash-Sutcliffe efficiencies (NSE; Nash and Sutcliffe [1970]) as well as coefficients of determinations ($R^2$), indicating that the hydrological model adequately captures the hydrology of the Tillamook watershed. Model calibration parameters derived at the Tillamook basin were then used for the Coos Bay basin. Both watersheds are similar hydrologically, defined by high winter flows driven by rainfall events, and low summer baseflows during the dry season. Therefore, it is expected that similar calibration coefficients should apply to both study sites.

After calibration and validation of the hydrologic models with observed data, simulations were then performed using CRCM input. For consistency with other aspects of the model framework, CRCM variables were all bias corrected using quantile mapping, Déqué [2007]) with the NARR dataset as the target. A full description of the utilized bias correction procedure, both the bivariate method utilized for wave modeling and the univariate method used for other variables, is beyond the scope of this paper. Instead, the reader is directed to Parker and Hill [2017] for a more detailed description.

The gridded hydrologic model produces a daily time series of streamflow at every coastal pour point along the coastline. However, the majority of these locations have small contributing areas and thus

produce very low flow values. Only the pour points with large contributing areas (watersheds) and streamflow rates were selected for inclusion in the hydrodynamic model. For Tillamook, data from the mouths of four rivers were included (the Kilchis, Wilson, Miami, and Trask) which were found to capture 95% of the basin annual streamflow. For Coos, seven points were chosen for inclusion
representing 90% of the basin annual streamflow. The streamflow from these points was aggregated into three inputs into the hydrodynamic model at the location of the Coos River, Palouse Slough and Noble Creek (Figure 3).

### 3.4 Monthly Anomaly Model

An important component of measured WLs along the west coast of the U.S. comes from Monthly Mean
Sea-Level Anomalies (MMSLA). These low frequency variations in WLs are caused by a wide variety of forcings ranging from local wind stress to large scale climate patterns (such as El Nino) [Chelton and Davis, 1982]. A comprehensive understanding of MMSLA remains complex, as they integrate a large number of simultaneous processes operating over a wide range of spatial and temporal scales. However, there have been many attempts to model the leading order terms found in MMSLA signals. This study
follows the work of Thompson et al. [2014] who used statistical regression and a subset of wind stress metrics to successfully reproduce the bulk of MMSLA variability across the west coast. The utilized regression is given by:

$$\eta = \eta_0 + a\tau_{eq} + b\tau_{ls} + c\tau_{xy} + \epsilon$$

$$(1)$$

where $\boldsymbol{\eta}$ is monthly mean sea level, $\boldsymbol{\eta}_0$ is the regression y-intercept term, $\boldsymbol{\tau}_{eq}$ is the equatorial wind stress, $\boldsymbol{\tau}_{ls}$ is the local alongshore wind stress, $\boldsymbol{\tau}_{xy}$ is the wind stress curl, $\epsilon$ is the residual error, and $(a, b, c)$ are regression coefficients. The reader is directed to the original source publication for additional information regarding the specifics of the wind stress coefficients as well as their scientific
basis.

Figure 4 shows the result of this regression for the Tillamook study site (Coos is not shown). While this formulation does not capture all variability in MMSLA ($R^2 \approx .6$), it does qualitatively capture the MMSLA signal and is based entirely on variables that are readily available from the NARCCAP
dataset. Furthermore, a statistical approach is attractive since directly modeling coastal MMSLAs would be very computationally expensive. The regression approach is found to somewhat underestimates extreme values of MMSLA which may introduce a low bias in calculated extreme total water levels. This bias should be similar for both the historic and future periods, so its effect on changes between those periods should be minimal. MSLAs are added to the hydrodynamic model time series as a post-
processing step (see Figure 2). Not including MMSLA within the hydrodynamic model may exclude some potential non-linear interactions between MMSLA, WLs and other forcings.

**3.5 Sea Level Rise**

SLR was included in the modeling framework as a change to mean sea level (MSL) within the hydrodynamic model. Projections were taken from the National Resource Council (NRC) report [NRC, 2012] which developed local estimates for SLR along the U.S. Pacific coast. These estimates include
contributions from steric/dynamic ocean modifications, glaciers and ice caps, sea-level fingerprint effects, and vertical land motion (e.g., isostatic adjustments). In calculating local SLR estimates, the NRC used a combination of IPCC 4th assessment projections (mid-range scenario) and an extrapolation methodology for the cryosphere components. This produced values larger than either the IPCC 4th or 5th assessments but still below some estimates for mean 2100 global SLR [Vermeer and Rahmstorf, 2009].

SLR data were taken from the nearest reported location to Coos and Tillamook Bay, Newport Oregon, which is situated approximately between the two study sites. The NRC report provides projection values (and ranges) for 2030, 2050, and 2100. A cubic spline was fitted to these values to allow a smooth interpolation to intermediate years. Multi-decade model runs of the hydrodynamic model were broken
into smaller 3-month periods and MSL was updated accordingly for each of these simulation blocks. This allowed for changes in MSL in a step-wise but nearly continuous fashion.

**3.6 Local Hydrodynamic/Wave Model**

The coupled ADCIRC-SWAN (ADCSWAN) model [Dietrich et al., 2011b] was used for this study. ADCSWAN is highly configurable in terms of implemented physics and readers are directed to the
source publication and model manuals for a full description of options and parameters. ADCIRC [Luettich and Westerink, 1992] solves the hydrodynamic portion of this pairing through the shallow-water equations. ADCIRC uses an unstructured horizontal grid which allows for finer spatial resolution in regions of complex bathymetry. Bathymetry for the model grid was developed through blending Oregon Department of Geology and Mineral Industries (DOGAMI) LiDAR [DOGAMI, 2009] and a
variety of National Oceanic and Atmospheric Administration DEMs [NOAA, 2018]. Wetting and drying were enabled due to the significant intertidal areas present in both bays. Non-linear bottom friction was used with a spatially variable friction factor set based on general land use classes [Dietrich et al., 2011a; Homer et al., 2011].

ADCIRC was run, for this study, in the 2D depth-integrated (2DDI) mode. Previous research has shown that storm surge and tidal signals can be accurately resolved by 2D barotropic models. Specifically, Resio and Westerink [2008] point out that 3D effects are readily absorbed by model calibration coefficients. Additionally, a sensitivity study by Weaver and Luettich [2010] found that differences in predicted WLs between 3D and 2D models were on the order of 5% over most of the domain. These
modest differences suggest that a 2D model can be an effective and efficient choice for studies of this type.

SWAN is a third-generation spectral model that solves the spectral action balance equation to compute the spectral evolution of wind waves. The unstructured format of SWAN [Zijlema, 2010] was utilized to allow tight coupling (on the same grid) with ADCIRC. SWAN was run in a nonstationary mode with offshore forcing provided by a temporally varying JONSWAP spectrum fitted to bulk wave parameters.

ADCSWAN was run with atmospheric forcing provided as gridded horizontal wind components and surface pressure; wave forcing using SWAN's nonstationary TPAR parametric spectrum file; hydrologic input as a normal flux into the domain, and tidal forcing at the ocean boundaries. Tidal forcing was defined as the eight locally dominant constituents (K1, O1, P1, Q1, M2, S2, N2, and K2) with location-dependent amplitudes and phases defined by the ENPAC tidal database [Mark et al., 2004] and the simulation time-dependent nodal factor and equilibrium argument defined by the T_tide harmonic analysis package [Pawlowicz et al., 2002]. The ADCSWAN model was first validated at each study site using a tidal simulation compared against NOAA tide gauge predictions. Using a one month simulation, both the Tillamook and Coos grids were found to have $R^2$ values greater than 0.97. The Tillamook model was additionally validated against the largest storm of record (the Great Coastal Gale of 2007; Crout et al. [2008]) with good agreements to extreme WLs [Cheng et al., 2015b].

## 4. Nonstationary Recurrence Intervals

This study primarily considers extremes which will be quantified in terms of RI events, since engineering design and community planning often rely on this concept. The traditional definition of an RI is built on an assumption of stationarity, or time-invariance. This assumption makes the definition of a RI simultaneously the inverse of the probability that an event of a given magnitude will be exceeded in a given year and the expected recurrence period of that event. This definition breaks down under nonstationary conditions, which can be experienced due to climate change and/or SLR. Reconciling a nonstationary environment with traditional design methods based on stationary assumptions is an ongoing challenge. Proposed alternatives include effective design value [Katz et al., 2002], expected waiting time [Olsen et al., 1998; Sala and Obeysekera, 2014], expected number of events [Parey et al., 2007, 2010], design exceedance probability, and design life level [Rootzen, 2013]. Each of these definitions represents a unique projection of the stationary case for nonstationary conditions. Problematically, the chosen metric can result in significantly different calculated RIs while most users simply interpret the result as comparable to the stationary case. This highlights the importance of rigorously defining utilized nonstationary RI formulation as well as considering if the utilized metric fits design conceptions.

Nonstationary extreme value analysis has recently seen a wide range of applications to coastal problems [Corbella and Stretch, 2012; Katz, 2013; Wahl and Chambers, 2016; Wahl et al., 2015]. Nonstationarity is generally incorporated within the statistical model by using time dependent parameters either as a linear or exponential function [Cheng et al., 2014; Ruggiero et al., 2010], a cyclical trigonometric function [Méndez et al., 2008; Mínguez et al., 2010] or as a more complicated function of covariates

[Méndez et al., 2007; Weisse et al., 2014]. Even for stationary extreme value analysis there is a range of commonly used statistical models with a corresponding uncertainty as a result of the chosen methodology [Wahl et al., 2017]. Across the wide variety of applications of nonstationary extreme value analysis, no consensus definition of nonstationary RIs nor a best-practice methodology has emerged.

This study approaches nonstationary RIs using the effective design value interpretation [Katz et al., 2002]. This defines a temporally varying RI (termed an effective RI, or design value, by Katz et al. [2002]) which holds the probability of occurrence for an event constant through time. This preserves an intuitive definition of RIs as well as how nonstationarity impacts extremes. Effective RIs add an additional dimension of time (in comparison to standard RIs) so are commonly presented as a family of curves with time on the x-axis, event magnitude on the y-axis, and each curve representing a recurrence interval (e.g. 2-year event). Additionally, this means that the specification of an effective RI requires both a recurrence interval and a time of interest (e.g. a 100 year event for 2050).

The effective RI definition of nonstationarity is chosen for this study due to the unique format of the results. WL data are output from the modelling framework in reference to MSL. Therefore, the WL time series does not show any discontinuity or trend from changing sea level, a signal that would only be visible if viewing WLs relative to a non-tidal datum. This results in an approximate stationarity, as a function of datum, and makes it possible to separate the calculation of RIs from the nonstationarity of the time series. In this context, calculating effective RIs reduces to calculating the stationary RIs and then adding SLR (the assumed nonstationary component) to these estimates.

The benefit to this approach is that it avoids the complications of fitting a nonstationary GEV and the corresponding loss of degrees of freedom from estimating the nonstationary trend from the data. Furthermore, most nonstationary GEV analyses are forced to use a priori simplistic functions due to limited degrees of freedom. This approach allows a more complicated trend that follows experienced SLR (approximately cubic for this study). The negative of the approach is the assumption/simplification that the resulting MSL timeseries is stationary. While this is a common statistical assumption, it does have consequences for the results and is discussed further in section 6.4.3.

### 5. Results

Model output was saved at a subset of model nodes (stations) in order to keep output files manageable in size. Output stations were spaced evenly across the bay in order to capture spatial variability (Figure 3). While ADCSWAN has the ability to write out numerous variables (including wave heights, periods, etc.) the focus of this study is on WLs so discussion here will be limited to that variable. The model was run in 3-month long segments with a 2 week overlap to avoid discontinuities in dynamic processes. Smaller segments were necessary for integrating SLR as well as for model stability reasons. Output data from these segments were then recombined into continuous time series at each station.

Analysis was performed for both the historic and future periods. With an identical configuration (for all modeling components) to the historic period simulation, the only free variable is the AOGCM forcing under climate change. Therefore, a comparison shows how extreme events can be expected to change (in a relative manner) over time.

### 5.1 Recurrence Intervals (Tide Gauge Locations)

RIs were calculated using a GEV distribution fitted to annual block maxima events. Figure 5 shows this analysis for observed, modeled historic and modeled future (without SLR included) time series at the Coos and Tillamook Bay tide gauge locations (Figure 1). The calculated historic period RI curve for Tillamook (Figure 5a) exhibits good agreement with observations with a maximum offset of around 4 cm. The agreement is less favorable for Coos Bay (Figure 5b), with a maximum offset of around 9 cm. For both locations, the largest difference between observed and modeled RIs is for medium (approximately 10-year) RI events. This is because both modeled RI curves exhibit a different curvature than the observed. A possible source in the low bias for modeled RIs may be the low bias observed in the MMSLA regression model.

It is important to note that Figure 5 includes future RI values plotted with SLR **not** included. This plot shows a comparison of the future return intervals (assumed stationary with SLR removed) to the stationary historic period return intervals. Since nonstationary and stationary return intervals are not generally equivalent, this provides a manner of comparison. Using the effective design value interpretation, this can be thought of as future RIs but with the chosen design year being 2000. Practically this shows future RIs as a function of only changing forcing (no SLR). Both plots show the future RI curve as exhibiting smaller WLs than the historic RI curve, hinting at a reduction in extreme WL forcing into the future.

Figure 5 presents results in a classic RI curve format but they can additionally be viewed as effective RIs (as described in Section 4). Effective RIs were developed by calculating stationary RIs from the WL time series (relative to MSL) and then adding SLR (Figure 6). Figure 6 contains both the future and historic effective RIs on the same plot to visualize how RIs change upon entering a nonstationary climate regime. Based off model assumptions, historic effective RIs are flat lines through time (as they include no nonstationary component). Figure 6 reiterates the same RI behavior as above with historic RIs being higher than future RIs for the current period (year 2000). Moving forward in time, this plot show that SLR eventually overtakes this effect to result in a higher future flood. The point of overtake (where nonstationary RIs begin to predict a larger flood) is plotted in Figure 6 as colored dots on the x-axis. This intersection is significantly earlier for short RIs (within 20 years for a 2-year event) than for longer RI events (over 60 years for a 100-year event). This result is shown for both estuaries but with Tillamook Bay having less spread in overtake time than Coos Bay.

**5.2 Recurrence Intervals (Spatial Variability)**

Section 5.1 discusses RIs at a single location (the tide gauge), but a key feature of this study's methodology is the ability to explore spatial variability in RIs across the study sites. Figure 7 demonstrates this by plotting the 100-year RI WL calculated for the historic period at each output
station. Analysis was limited to stations that were wet over 75% of the record length. This was implemented to limit uncertainty in GEV analysis due to insufficient record lengths at only periodically wet stations.

It is apparent in Figure 7 that there is a significant gradient in extreme WLs across both study site
estuaries. For Tillamook, WLs differ by approximately 25 cm and for Coos by around 35 cm. The gradient in WLs is orientated such that that the minimum WL is located near the estuary entrance with extreme WL height increasing with further distance into the estuary. Of particular interest, this trend is the same for both study sites suggesting that this pattern may be more generally applicable. While Figure 7 only plots the 100-year event, this extreme WL differential is maintained across other RI
periods.

The spatially variable WLs produced from this analysis provide the necessary information for building flood maps. This was accomplished by fitting a smooth surface to the scattered stations using spatial spline interpolation [ESRI, 2016]. The 100-year RI WL surface was then intersected with the estuary
DEM with all locations below the extreme WL surface defined as "flooded". This methodology makes the assumption of extrapolation at the edge of the surface where station output was not available. The calculated flood surface is shown in Figure 7 as a light blue surface. Comparison of this surface to the U.S. Federal Emergency Management Agency (FEMA) 100-year flood plain for Coos Bay [FEMA, 2014] (not shown) revealed that the produced flood inundation zones are different, but not greatly so.
This is likely more attributable to Coos bay's steep topography than to similarities in produced hazard levels. Planform flood area is highly influenced by terrain gradients and large variabilities in predicted hazards can manifest as only small changes to flood zone for steep shorelines. FEMA only produces flood map products, so a more quantitative comparison of extreme water levels was not possible.

**5.3 Changes to Recurrence Interval Spatial Structure**

Section 5.2 details the importance of considering spatial variability in extreme WLs but only considers the historic scenario. An important question remains as to how this spatial variability can be expected to change moving into the future. This is especially important as, while individual flooding estimates might be biased (for example by the inability of the MMSLA regression to produce extremes or by bias in the forcing AOGCMs), these biases should cancel when calculating change from the historic to future
scenarios. Figure 8 explores this analysis by plotting the difference between future effective 100-year RI WLs (calculated for the year 2050) and historic 100-year RI WLs.

As a primer, if the RI difference plot (Figure 8) showed no spatial variability, then the same pattern seen in Figure 7 would be replicated in the future, with only a vertical offset. This would be an important result signifying that the spatial pattern of current extreme flooding events will remain unchanged into the future with only an estuary-wide SLR adjustment being required to update hazard assessments.

Instead, Figure 8 shows a spatial pattern that is qualitatively similar to that seen in the historic RI plots. Specifically, results show that the change in WLs increases with further distance into the estuary, a result that signals an increase in the spatial gradient of RIs as the study sites move into the future. Looking at other RI event periods (not shown), results show a gradual decrease in changes to spatial variability from around 10 -20 cm for the 100-year RI event to approximately no change for the 2-year

RI event. Note that the choice to plot effective return intervals for the year 2050 is arbitrary and does not affect the spatial pattern of differences (the critical information in this plot). A different design year would only change the overall magnitude, not the inter-point variability / pattern. Also included in Figure 8 is the 100-year recurrence interval flood zone for the historic period and for the effective RI design year of 2100.

**6. Discussion**

**6.1 Drivers of Extreme Water Levels**

The spatial variability of extreme WLs shown in Figure 7 motivates further investigation into what physical processes are controlling WLs in different regions of the study areas. This was investigated by carrying out a series of 17 day simulations bracketing the largest observed event at each study site and

for each time period. Each simulation in the series was performed with an individual forcing component turned off (e.g., wind, streamflow, etc.). The WL contribution from each forcing was then calculated by subtracting the simulation with the forcing of interest turned off from a simulation with full forcing. Therefore, each component can be conceptually thought of as quantifying the contribution of each forcing of interest to the total maximum WL. Figure 9 summarizes the breakdown of contributions at

several locations (moving up the estuary) in each study site, and for both historic and future periods. Components are plotted at the time of maximum WL occurrence.

The results show that, for the simulated events, pressure and MMSLA are the largest components of non-tidal residual (tides are not shown in Figure 9). As MMSLA are a source of uncertainty in this

study's modeling framework, this motivates a need for further improvement of MMSLA estimates. This could potentially be accomplished either through improved statistical methods or a computationally tractable physical modeling approach. Streamflow and offshore wave forcing were found to be an order of magnitude less important than pressure and MMSLA. This is especially true for the two events at the Tillamook study site (Figure 9, subplot c and d) where streamflow and wave forcing were found to be of

negligible importance to extreme event WLs. The wind contribution was found to be variable across stations and events. This is expected as wind setup is highly dependent on the estuary geometry and the wind direction of the specific event.

However, this analysis is only for the maximum annual event and other events likely have different compositions in terms of forcing contributions. Further investigation shows that, for both study sites and both climatological periods, extreme event magnitude is not correlated with any individual forcing (p-value > 0.05 when calculating correlation between WL magnitude and forcing magnitude at annual maximum events). This is reinforced by the fact that many annual maximum events are found to occur during below average non-tidal forcing conditions (e.g. below average wind or waves). This supports the conclusion that extreme WLs in the PNW are generally compound events, driven by the sum of multiple forcing that are not necessarily extreme themselves. This result is in agreement with other studies of forcing contributions to extreme events in the PNW [Parker, 2019].

The exceptions to this conclusion are tides and MMSLA, which are found to have a statistically significant correlation with event magnitude. Tides were not plotted in Figure 9 for scale reasons but were found to be the largest fraction of WLs (an average of 185 cm for Till and 145 cm for Coos). It follows that extreme WLs would most often occur during (or near) a high tide. Therefore, the concurrent timing of tides and non-tidal forcing becomes a major control on WL magnitude. This represents a mechanism explaining the predominance of compound events in the PNW, in which extreme WLs are not necessarily associated with extreme forcing. This is also a potential reason for some forcing components showing a negligible influence to extreme WLs (e.g. waves in Figure 9). While waves have been shown to be important drivers of non-tidal residual in PNW estuaries [Cheng et al., 2015b; Olabarrieta et al., 2011], tidal modulation means extreme WLs do not necessarily occur during maximum wave energy events.

The resulting evidence of complexity in compound events for the study site confirms that a comprehensive analysis of extreme WLs in the PNW likely requires an approach similar to that taken here. Event based approaches would likely be ineffective as storms or extreme forcings are not necessarily correlated with max annual events. Additionally, the common methodology of simply adding the largest non-tidal residual to a high tide could result in significant overestimations of event magnitude.

## 6.2 Spatial variability in Return Intervals

It is common practice to calculate WLs at a convenient location (such as a tide gauge) and then apply this value across the entire study domain. While larger estuaries (e.g., Delaware Bay) may have multiple tide gauges, smaller estuaries typical of the PNW tend to have one or none. A spatially constant assumption represents a major simplification as even tides can produce significant spatial WL variability in semi-enclosed basins [Holleman and Stacey, 2014]. Additionally, spatial variability is particularly important for estuaries as they are often regions of low-gradient topography where a modest change in water elevation can correspond to a large change in inundated area. Results (Figure 7) show variability in WLs for both locations in excess of 25 cm. Furthermore, the smallest WLs are at the estuary mouth where, in the PNW, the tide gauge is generally located. This means that estimating

flooding from a tide gauge will result in under-predictions for flooding with errors increasing with upstream distance into the estuary. This result strongly supports the importance of considering spatial variability in WLs within flood hazard assessments.

Between the two study sites, Coos is found to have a larger extreme WL differential (approximately 30 cm from the mouth to the interior bay). This is contrary to the expected result that Tillamook, with its proportionally larger forcing, would have a larger gradient. Streamflow, in particular, was expected to have a significant impact on WLs but was found to have a minimal effect for both study sites. This was particularly true for Tillamook despite its larger average streamflow input. Figure 9 shows that, while

the streamflow component does increase moving shoreward for Coos bay, the majority of the WL differential for both sites is driven by pressure. An additional component of the WL gradient is from tidal forcing which produces around a 10 cm differential between the estuary mouth and the inner bay for both locations.

Results show that spatial variability is predicted to change into the future. However, this result is primarily shown for longer RI events with shorter RI events not showing any change in spatial variability for the future scenario. Shorter RI period events are better constrained statistically than longer RI period events so it is possible that some proportion of the predicted spatial variability is a function of GEV analysis on a temporally limited record. A modeled record longer than the period used

here (20 years for historic, 30 years for future) could help to illuminate if this conclusion is a physical result.

### 6.3 Changing Extreme Events

With analysis suggesting RIs evolve through time, a natural next question is how climate change is modifying the estuarine system. To help illustrate this, Table 2 shows a comparison of forcing statistics

under the historic and future climatological periods for Tillamook. Results show a decrease in most forcing variables, both as an overall average and for average forcing during extreme events. This result is similarly seen for Coos Bay (not shown). This suggests that the decrease in RI shown in Figure 5 is caused by a broad scale reduction of forcing on the estuary for the GCM scenario considered. Unfortunately, with all modeled forcing shown to be reduced for the future period, it is difficult to

conclusively differentiate which drivers are controlling changing RIs. Similarly, as GEV analysis is based on a parametric fit of multiple annual maximum events, it is difficult to characterize the cause of changing RIs without considering the aggregate behavior of all the events to which the GEV is fitted (rather than a single event as shown in Figure 9).

A further exploration of changing RI was shown through effective RIs (Figure 6). This result found that the reduction in forcing is eventually overcome by SLR, although with the timing being controlled by the size of the RI event. The idea of an "overtake point" is a simplification based on the assumptions within the statistical model, specifically that of stationarity/nonstationarity (see Section 6.4.3). Another way of viewing this result is that SLR represents a single value for change across all RIs. By year the

2050, both 2-year and 100-year RI events increase by 17 cm due to SLR. On the other hand, the change in RI magnitude from forcing is variable across return periods. From forcing, the change in 2-year RI is only 3 cm while for the 100 year RI it is much larger at 32 cm. This means that shorter RI events are less buffered by a change in forcing than longer RI events. The conclusion is the same from this interpretation in that short RI events will be comparatively more impacted by SLR than longer RI events.

## 6.4 Modeling limitations

### 6.4.1 Excluded Processes

In this study bathymetry/topography was held constant through time. Bathymetry is a first order control on flooding and so an ideal future projection would include morphological evolution of the estuary. This said, morphological projections at climate change scales are extremely uncertain. The combination of high uncertainty and high dependence would leave resulting flood predications dominated by an uncertain morphology projection with all other signals obscured. This study therefore does not consider morphological evolution in order to specifically highlight how changing forcing impacts extreme WLs.

Both estuaries have significant anthropological modifications ranging from coastal infrastructure to dredged channels. Coos in particular has an engineered coastline along the majority of its southern boundary. A key factor in future extreme events is the interaction between human intervention and the estuarine system. For example, estuary WL characteristics under tidal forcing show high sensitivity to anthropological changes [Gallien et al., 2011; Wang et al., 2017] and modifications to land use have been shown, in certain cases, to be of the same order of importance to WLs as SLR [Bilskie et al., 2014]. Dredging, for the same reason as morphological evolution, can cause drastic changes to estuarine hydrodynamics. However, similarly to morphological predictions, anthropological controls are highly uncertain and therefore not included in this analysis.

### 6.4.2 Climate Model Variability

The results shown in Figure 5 provide an interesting comparison of modeled and observed extremes. However, the modeled results and observed results are not based on the same forcing time series, but rather one observational and one modeled time series. Since both the Coos and Tillamook modeled RI curves have less curvature than the observed curves, this could be a result of the specific climate model iteration that was used for the simulation. The common solution to this problem is the usage of ensembles of climate models rather than a single AOGCM [Murphey et al., 2004]. Unfortunately running ensembles is computationally expensive given the level of complexity included in this study. Conceptually this study makes the compromise of including more physical complexity at the cost of uncertainty quantification. Results from this project should therefore not be viewed as a probabilistic or a "most-likely" result from climate change. Instead, they should be thought of as a single possibility of what could happen and an illustration of the importance of including various processes in a study of this type.

Not including multiple model iterations is also problematic in constraining the current climate's RIs. Results in Section 6.1 show the importance of compound events and forcing timing (especially tides) at these study sites. While forcing was bias corrected using a methodology that has been shown to perform well for extreme quantiles [Parker and Hill, 2017], timing of forcing occurring on a high tide or during a high MMSLA is additionally critical. This study examines only one possible combination of forcing timings that may or may not be representative of the overall extreme behavior of the system. This could once again be addressed through usage of ensembles or multiple iterations of the current climate.

While results from this study build a strong case that including dynamic coupling of processes is important for flood estimation, a natural next question is the cost/benefit when viewed under the extreme uncertainty of climate change. As an example, this study shows a change in spatial distribution of extreme water levels of 10-20 cm moving into the future. This is significant in terms of hazard quantification but small in comparison to uncertainty in PNW sea level rise by 2100 (on the scale of over 60 cm: Miller, 2018). It is expected that the need to include this uncertainty will likely often preclude the usage of the coupled dynamics employed by this study. Rather it is hoped that these results can provide an idea of the type of errors being induced by using more simplified modeling frameworks. Furthermore, this study provides a strong motivation for methodologies to combine dynamic modeling with faster simulation times. Recent research at a similar estuary study site in the PNW has shown emulation as a promising method to provide this linkage (Parker et al., 2019).

### 6.4.3 Assumptions of Stationarity

Another important assumption in this study is that of stationarity when SLR is removed. The reality is somewhat more complicated as climate change will result in a forcing driven nonstationarity in addition to that seen from SLR. Research has additionally shown that SLR can be expected to interact non-linearly with storm surge, creating another source of nonstationarity over time (Buchanan et al., 2017; Devlin et al., 2017; Wahl, 2017). For our case, each time series segment is statistically stationary (Augmented Dickey–Fuller test, p-value < 0.001) but the overall time series (from 2000 to 2070) must be nonstationary. This is because the two segments (historic and future) show distinctly different calculated RI curves (Figure 5) so WL behavior, as controlled by forcing, must be changing. Simply put, this study analyzes two segments that are not long enough to reveal the longer term nonstationary behavior of the overall time series as controlled by changing forcing. This is not problematic for this study, which compares two snapshots, but a comprehensive analysis would need to resolve the overall nonstationarity from forcing as well as from SLR. This is also a factor in the calculation of the overlap timing from effective RIs (Figure 6) since the overall nonstationarity from forcing is not included in the analysis. Therefore, the overlap timing results should not be considered an exact calculation but rather a general result.

### 7. Conclusions

This paper introduced a process-based modeling framework for analyzing climate change impacts to various controls on estuarine flooding. In particular this study focused on extremes and changes to RI events at two PNW estuaries. This study described the difficulty of using RIs in the context of nonstationarity and showed how "effective RIs" can be an intuitive way of understanding changing flood hazards. Effective RIs showed that predicted changes to forcing result in a decrease in extreme event magnitude moving into the future. This decrease buffers the increase in WLs that comes from SLR. This buffering effect was shown to be smaller for short RI events than long RI events suggesting that increasing extremes will be felt first for low RI events.

This study used multiple study sites and climatological periods to explore drivers of extreme events. It was found that extreme events for both locations were not controlled by a single forcing but rather by compound events. Tides were shown to be the largest contributor to extreme WLs. The requirement for a high (or near high) tide modulates the contribution from other forcing during extreme WLs. This means that high non-tidal residual or storms are not necessary the source of extremes at the study site. This suggests that both event based methodologies and the common procedure of adding an uncoupled high tide and high non-tidal residual will both result in an incorrect assessment of flooding magnitude.

An additional outcome of this study was the demonstration that extreme WLs are spatially variable in estuaries. The results showed that WLs varied by more than 25 cm across each estuary domain. Relying only on predictions at the tide gauge and the assumption of a horizontal water surface will therefore mischaracterize flood risk. Since this study found that WL gradients for long RI events increased in the future, errors associated with 'bathtub' approximations of flooding surfaces will similarly increase. Overall this study highlighted the importance of limiting conclusions drawn from point (tide gauge) analysis to regions spatially near observations or to rigorously define uncertainty from not sampling the full spatial variability of flood surfaces.

### Acknowledgments

The utilized NARCCAP climate data are open access and available through the Earth System Grid climate data gateway. The NARR climate dataset is available through NOAA's Earth System Research Laboratory website. Tide gauge records are available through the National Oceanic and Atmospheric Administration (NOAA) National Ocean Service (NOS) website. River discharge is available from the USGS through the National Water Information System. Wave buoy information was obtained through NOAA's National Data Buoy Center website.

This work used the Extreme Science and Engineering Discovery Environment (XSEDE), which is supported by National Science Foundation grant number ACI-1548562. Resources were provided by the

XSEDE STAMPEDE cluster at the Texas Advanced Computing Center through a series of allocations. We thank the XSEDE program for providing computational resources that made this project possible.

This paper was funded in part by Oregon Sea Grant under award (grant) number NA14OAR4170064
(CFDA No. 11.417) (project number R/CNH-25) from the National Oceanic and Atmospheric Administration's National Sea Grant College Program, U.S. Department of Commerce, and by appropriations made by the Oregon State Legislature. Additional funding was from the Oregon Sea Grant Robert E. Malouf marine studies scholarship (Project number E/INT-143). The statements, findings, conclusions, and recommendations are those of the authors and do not necessarily reflect the
views of these funders.

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

**Figures**

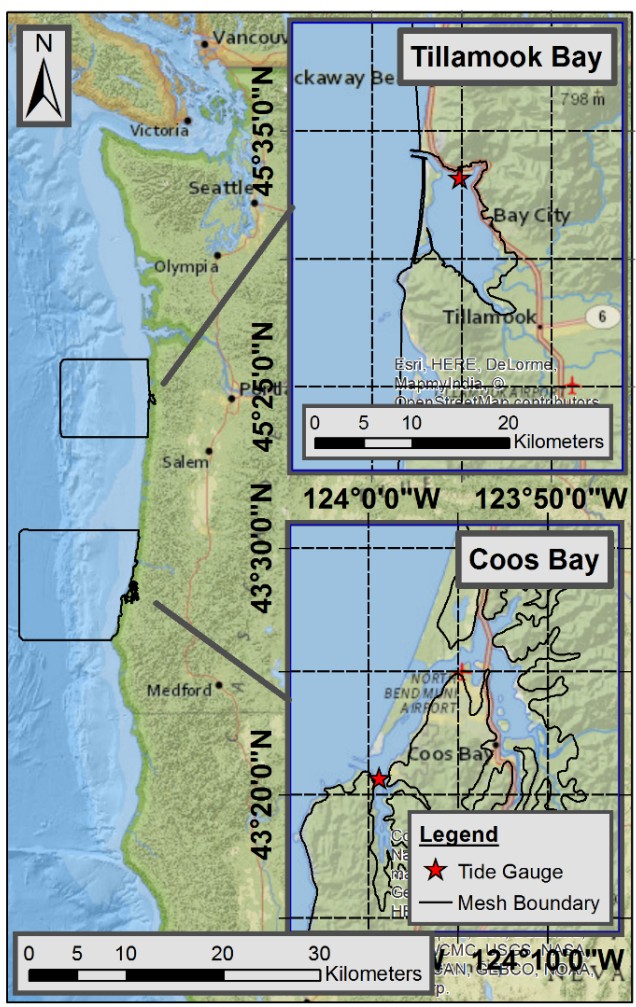

Figure 1: Overview map (left) and detail views (inset boxes) of Tillamook and Coos Bays on the U.S. Pacific Northwest coastline. Both insets are at the same scale (shown on Tillamook Bay inset). The Tide Gauge locations are shown as red stars and the mesh boundary is shown as a dark line on both the overview and detail view boxes.

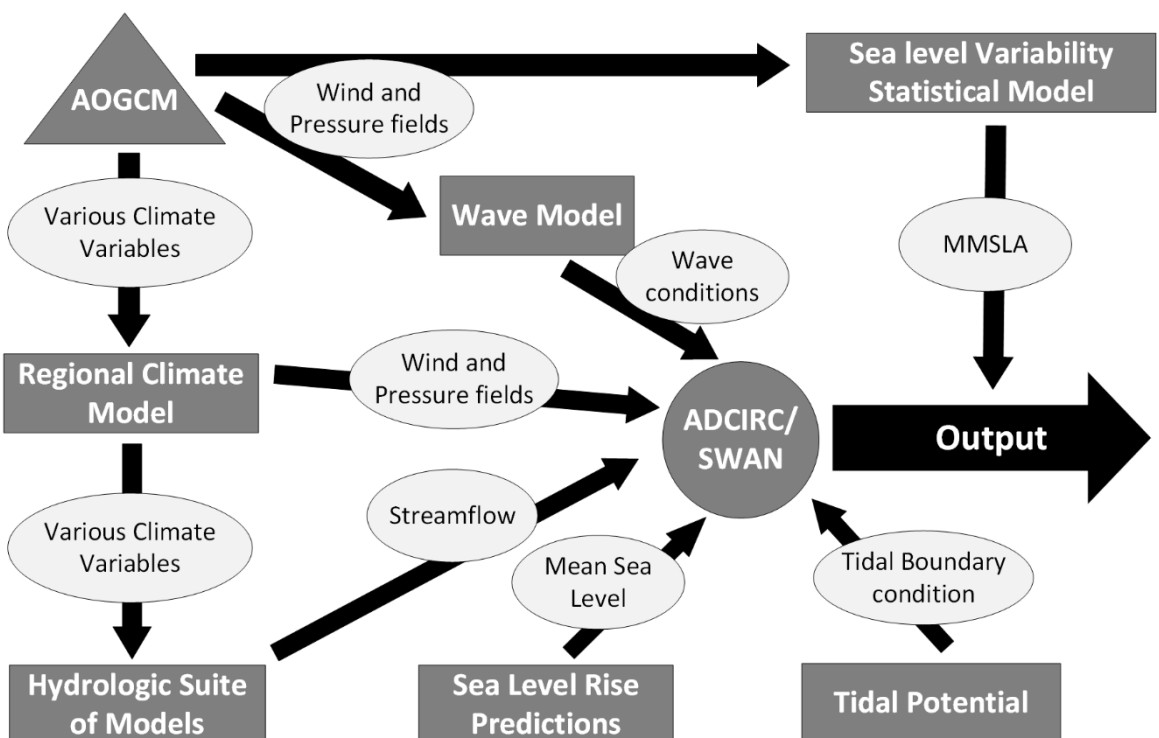

**Figure 2: Overview of the general modeling framework. The dark gray triangle labeled AOGCM in the top left corner is the "parent" model. Dark gray rectangles represent sub-models. Light gray ovals represent variables that are passed between modeling components.**

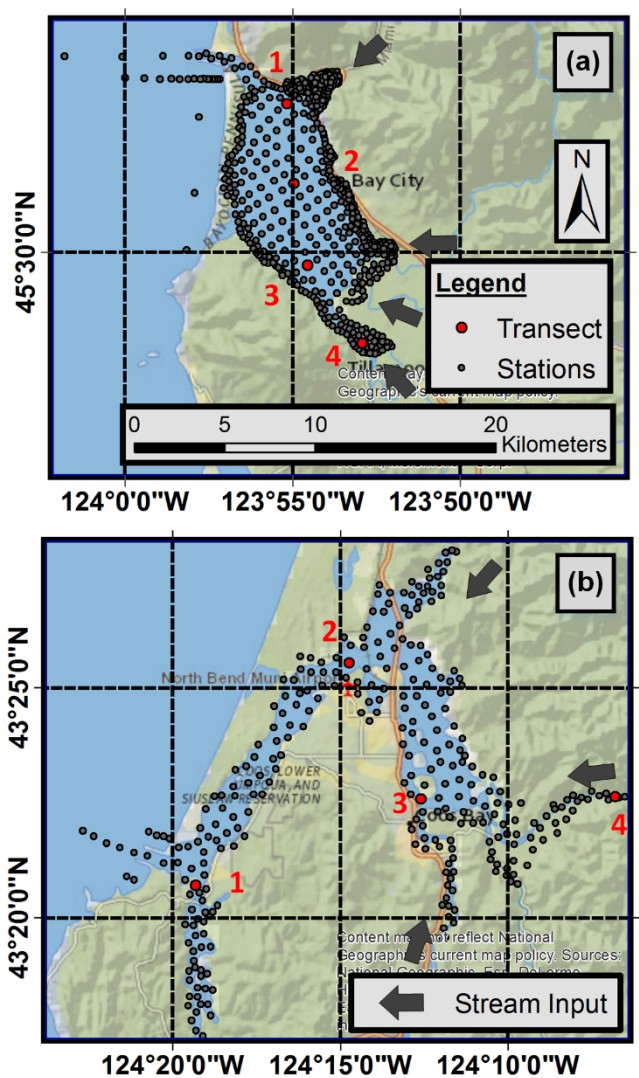

**Figure 3: Station output locations at the Tillamook and Coos Bay study sites, panel (a) and panel (b) respectively. Grey arrows show hydrologic inputs into the model domain. Red points and numbers represent transect station locations (see Section 6.1). Both figures are at the same scale (shown in panel a).**

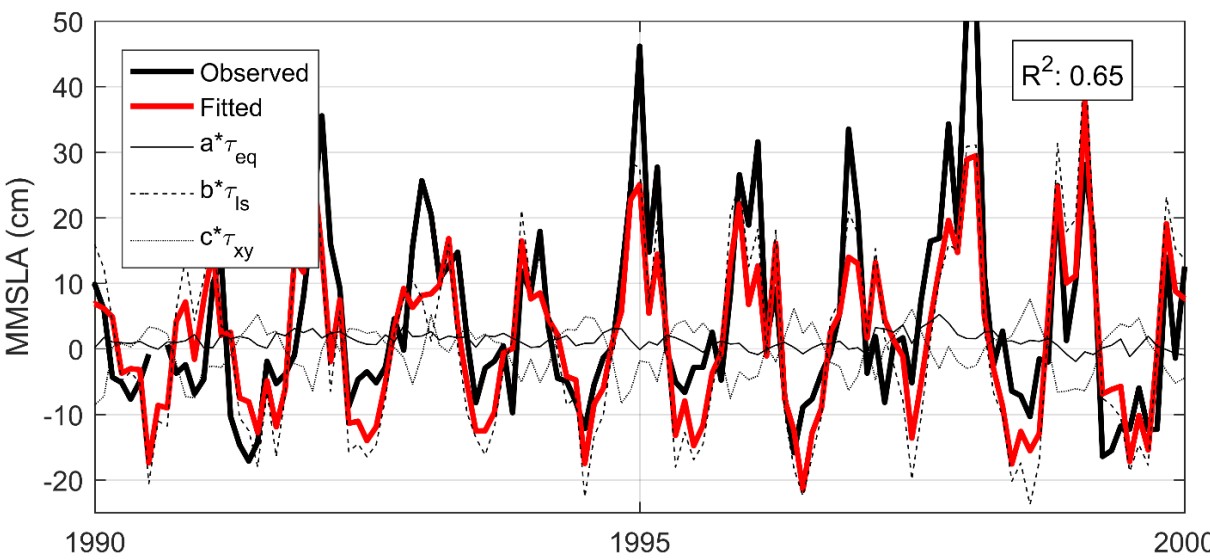

**Figure 4: MMSLA regression for the Tillamook study site in the style of Thompson et al. [2014]. Fitted contributions to MMSLA from predictor variables $\tau$eq, $\tau$ls, and $\tau$xy are shown as thin black lines (full, dotted and dashed respectively). The bold black line is the observed MMSLA signal while the bold red line is the total fitted MMSLA signal.**

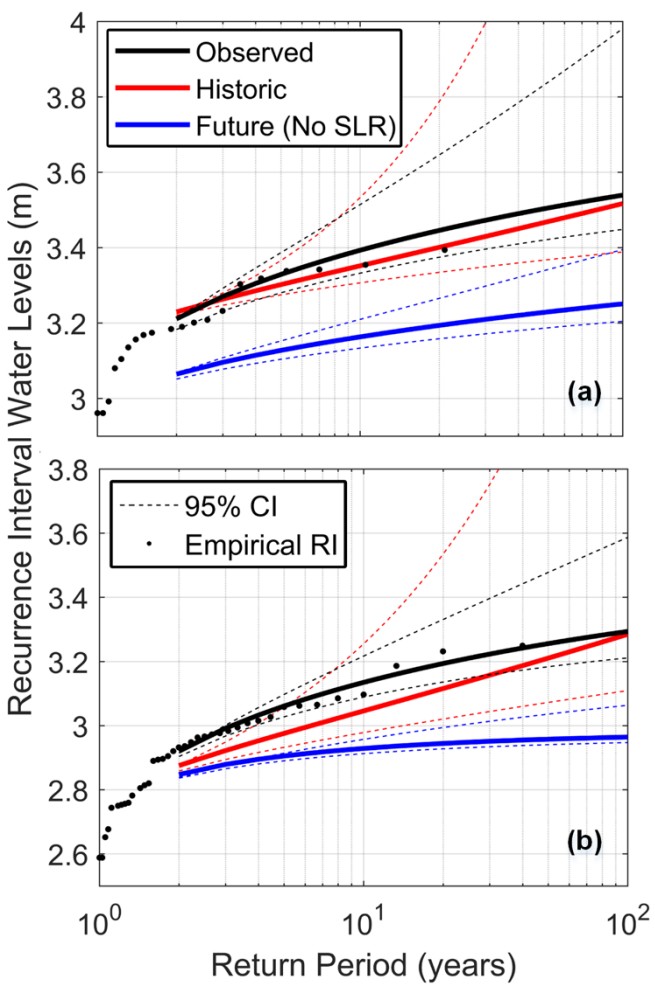

**Figure 5: RIs for the Tillamook and Coos Bay tide gauges, panel (a) and panel (b) respectively. WLs are in the NAVD88 datum. Confidence intervals are only for statistical uncertainty in the GEV model and are calculated using a likelihood-based method (plotted as dotted lines). SLR has not been included for future RI curves.**

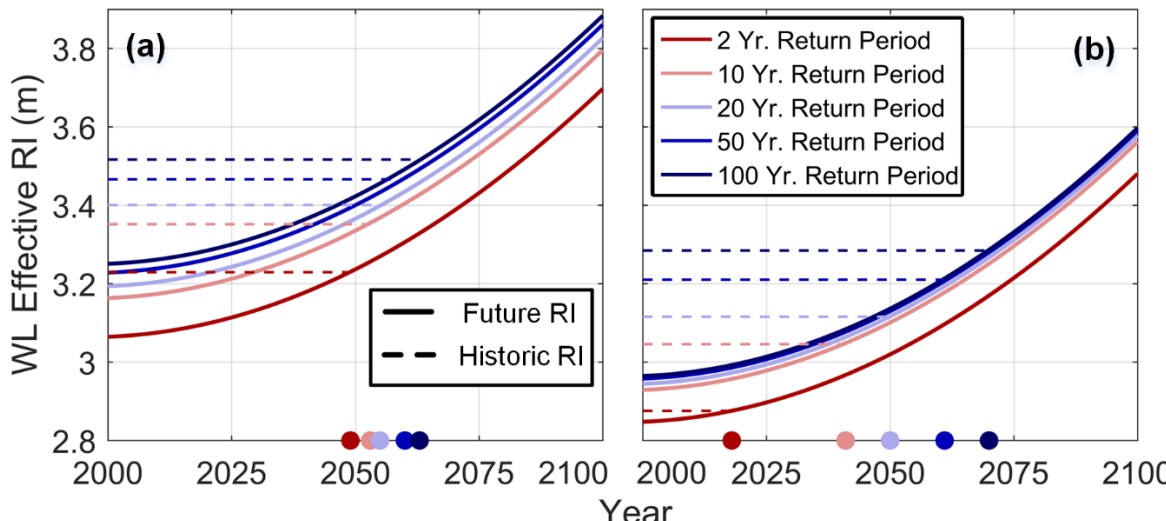

**Figure 6: Tillamook (a) and Coos Bay (b) effective RI WLs for the historic (dashed line) and future (solid line) periods. Shorter return periods are represented as hot colors while longer return periods are plotted as cool colors. The location of intersection between the historic and future effective RI curves is plotted on the x- axis as a dot with color corresponding to return period. The x- and y- axis scaling is the same in each subplot.**

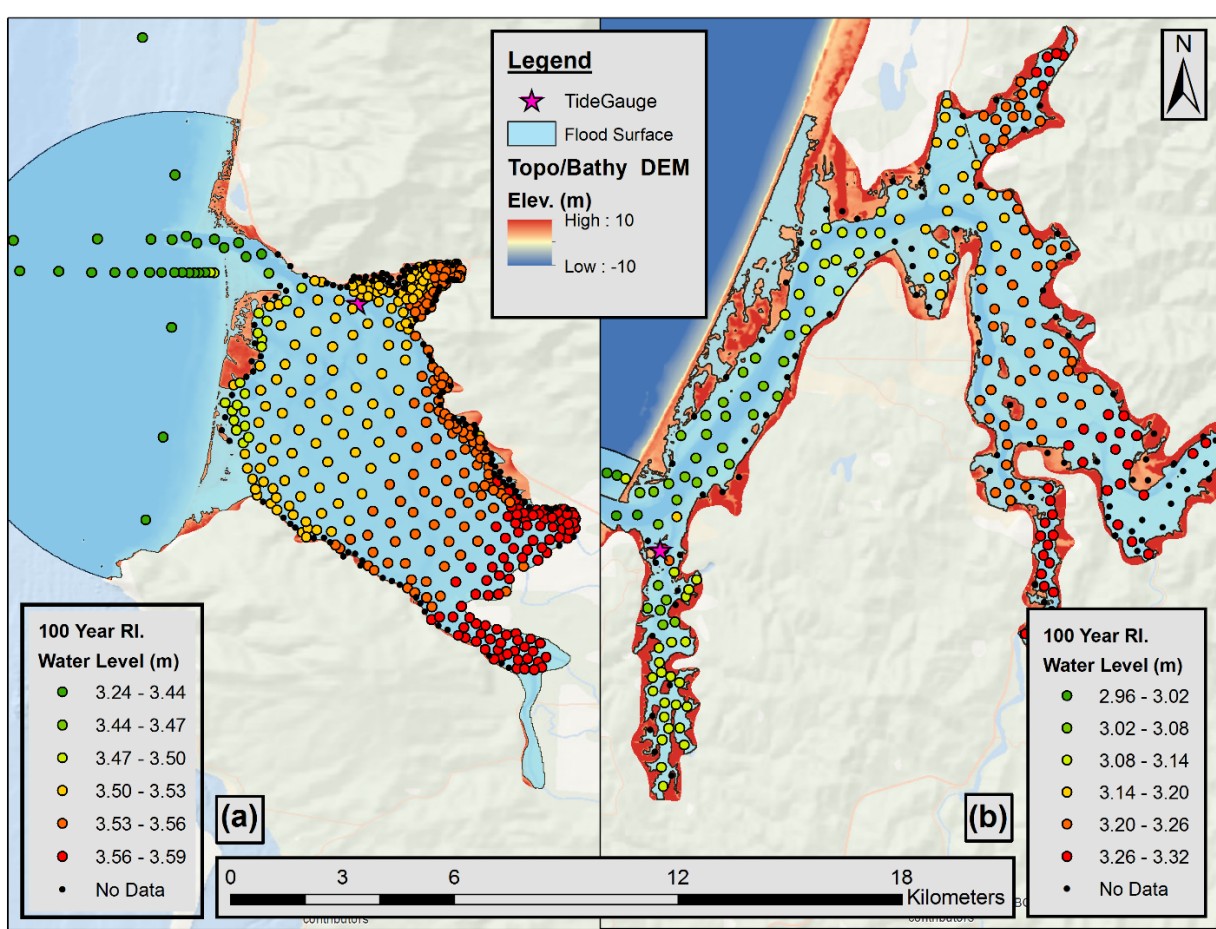

**Figure 7: Historic period 100-year flood surface results for Tillamook (a) and Coos (b) Bays. Individual stations are plotted with color scale indicating the historic period modelled 100-year RI WL magnitude. Note that the color scale is different between the two study sites. The calculated flood inundation surface is plotted as a blue transparent region.**

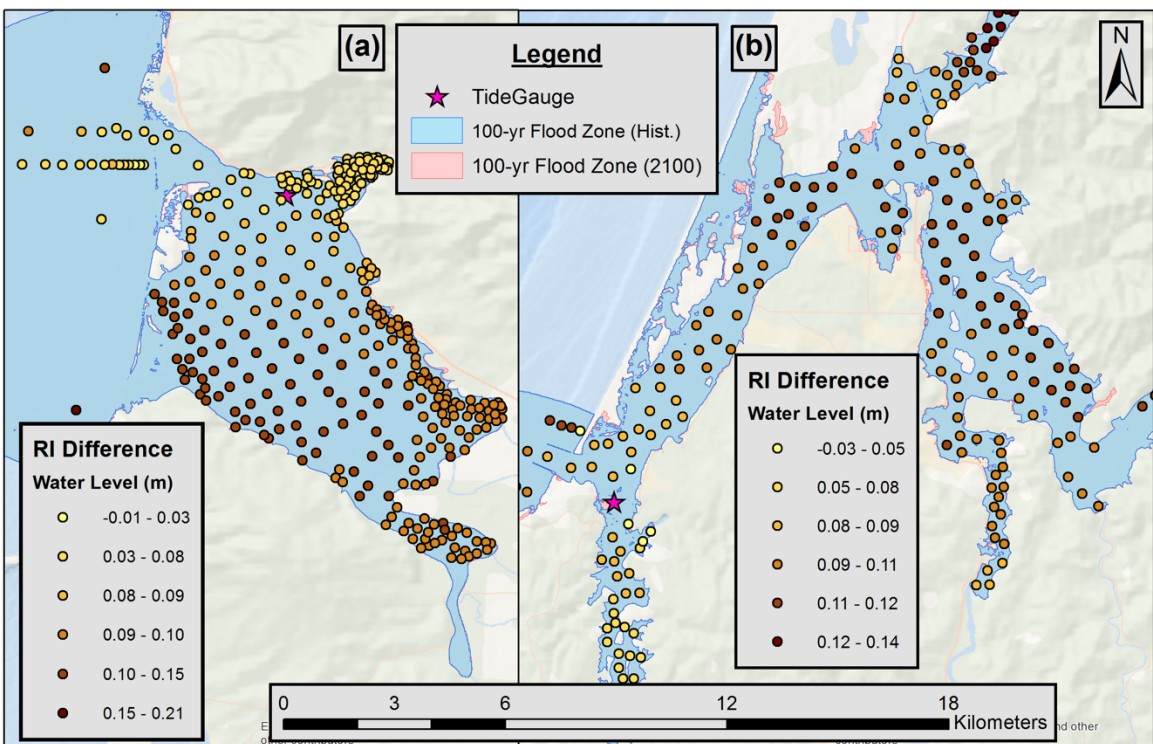

**Figure 8: Changes to 100-year RIs plotted at output stations. Changes are calculated as the future effective RI WL (at year 2050, 0.172 m of SLR) minus the historic RI WL. The blue surface is the calculated historic 100-year floodplain while the red surface is the future 100-year flood surface calculated for the year 2100.**

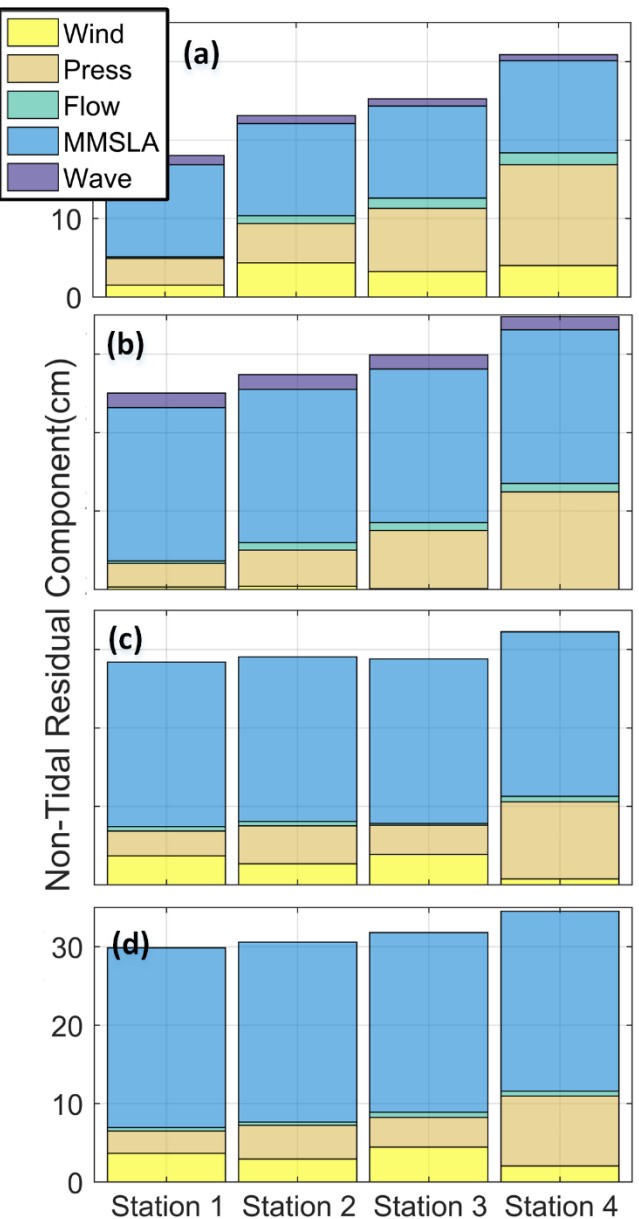

**Figure 9: WL contributions from various forcings during the largest simulated WL event. Stations transition from the tide gauge (station 1) to the far river outlet of the estuary (see Figure 3 for station locations). Panel (a) and (b) are for the Coos historic and future periods. Panel (c) and (d) are for the Tillamook historic and future periods. All panels have the same y-axis scaling.**

**Tables**

Table 1: Coos and Tillamook estuarine characteristics (after Engle et al. [2007]). Wind characteristics are at tide gauge locations. Wave characteristics are offshore at buoys 46002 and 46005. Streamflow values are from this study's hydrological analysis

| | Coos | Till. | |
|---|---|---|---|
| Estuary Area | 43.8 | 37.9 | $km^2$ |
| Estuary Drainage Area | 1520 | 1430 | $km^2$ |
| Estuary Volume | 0.207 | 0.071 | $m^3x10^9$ |
| Ave. Daily Flow | 61 | 85 | $m^3/s$ |
| Tidal Prism | 0.066 | 0.061 | $m^3x10^9$ |
| Mean Tidal Range | 1.7 | 1.9 | m |
| Ave Summer Salinity | 25.8 | 24.5 | psu |
| Ave. Depth | 3.5 | 1.4 | m |
| Ave. Offshore Wave Height | 2.7 | 2.8 | m |
| Ave. Wind Speed | 2.4 | 4.6 | m/s |
| Ave. Wind Direction | 164 | 190 | deg. |

**Table 2: Comparison of forcing for Tillamook Bay under the historic and future climatological periods. "Ave. Overall" is a full time series average while "Ave. Annual Max" is an average of forcing during observed annual WL maximum events.**

| | WL (m) | Hs (m) | Flow (m³/s) | Wind Mag. (m/s) | Wind Dir. (deg.) | Press. (Pa.) | MMSLA (cm) |
|---|---|---|---|---|---|---|---|
| **Historic Period** | | | | | | | |
| Ave. Overall | 0.01 | 2.8 | 85 | 4.5 | 340 | 98,400 | 0 |
| Std. Overall | 0.82 | 1.4 | 82 | 2.6 | 289 | 660 | 11 |
| Ave. Annual Max | 2.01 | 4.5 | 172 | 7.5 | 283 | 97,800 | 17 |
| **Future Period** | | | | | | | |
| Ave. Overall | 0.01 | 2.6 | 85 | 4.5 | 339 | 98,400 | 0 |
| Std. Overall | 0.83 | 1.4 | 89 | 2.6 | 287 | 630 | 11 |
| Ave. Annual Max | 1.99 | 3.8 | 111 | 6.2 | 344 | 98,300 | 15 |

**Author Contribution**

Kai Parker and David Hill designed the modeling framework. Gabriel García-Medina performed the Wavewatch III simulations. Jordan Beamer performed the hydrologic simulations. Kai Parker
10  performed the hydrodynamic simulations and data analysis of results. Kai Parker prepared the manuscript with contributions from all co-authors.

**Competing Interests**

The authors declare that they have no conflict of interest

**Data Availability**

15  Data and Code can be made available upon request.