# Peer review of "The Effects of Changing Climate on Estuarine Water Levels: A United States Pacific Northwest Case Study"

_Natural Hazards and Earth System Sciences, 2018_

## Referee Comment (RC1) · Anonymous Referee #1 · 5 Feb 2019

The authors have proposed a modeling framework to quantify the extreme estuarine water levels (WL) under climate change. Their framework, taking oceanic, atmospheric and hydrologic processes into account estimates extreme water levels in two estuaries of US Pacific Northwest. The idea of integrated modeling that couples processes across scales to estimate extreme estuarine water levels is interesting, and I believe this idea deserves publishing in NHESS, however after a major revision. There are a few issues with this version that I explain below:

Major: - First one, which the authors themselves have briefly mentioned, the climate and sea level rise projections used here are relatively old-dated! Publishing research

[Figure]

based on 4th IPCC assessment report, while IPCC 5th has been around for years and IPCC 6th is coming out soon, needs a valid justification. To me, saying "was the only climate product, at the start of this project" is not enough. I expect a good justification that the current results are still useful for the audience.

- At the end of section 4 (Lines 1-9 in Page 9), the non-linear interactions between SLR and tide/surge (i.e. Wahl, 2017, Sea-level rise and storm surges, relationship status: complicated!, Environ. Res. Lett., https://doi.org/10.1088/1748-9326/aa8eba ; Devlin et al., 2017, Coupling of sea level and tidal range changes, with implications for future water levels, Scientific Reports, vol 7, 17021) should be highlighted that are missed here. Also, other alternatives for non-stationary frequency analysis should be explained as well for interested readers (i.e. Cheng L., AghaKouchak A., Gilleland E., Katz R.W., 2014, Non-stationary Extreme Value Analysis in a Changing Climate, Climatic Change, 127(2), 353-369, doi: 10.1007/s10584-014-1254-5.).

- Page 4, L9-10: the hindcast and forecast periods are not the same length. This makes them incomparable, right?

Minor: Page 1, L 31-32: There are some studies to be cited here, i.e. Jay et al., 2016, Tidal-Fluvial and Estuarine Processes in the Lower Columbia River: II. Water Level Models, Floodplain Wetland Inundation, and System Zones, Estuaries and Coasts, Volume 39, Issue 5, pp 1299–1324; and the references therein.

Page 2, L18: a relevant recent citation Gallien, et al., 2018, Coastal Flood Modeling Challenges in Defended Urban Backshores, Geosciences 2018, 8(12), 450; https://doi.org/10.3390/geosciences8120450.

Page 6, L4: Please provide more details about the quantile mapping technique used here.

Best of luck,

2018-383, 2019.

---

## Author Comment (AC1) · 27 Feb 2019

Response to Anonymous Referee #1 Recommendation: Major revision Summary:

"The authors have proposed a modeling framework to quantify the extreme estuarine water levels (WL) under climate change. Their framework, taking oceanic, atmospheric and hydrologic processes into account estimates extreme water levels in two estuaries of US Pacific Northwest. The idea of integrated modeling that couples processes

across scales to estimate extreme estuarine water levels is interesting, and I believe this idea deserves publishing in NHESS, however after a major revision. There are a few issues with this version that I explain below: "

Response: We appreciate the reviewer's assessment of the paper and especially their detailed suggestions as to how to improve the manuscript. We have carefully incorporated all the reviewers comments and our responses can be seen in the following text. Our comments have additionally been attached as a pdf for better readability. We hope that, with the reviewer's comments integrated into the new manuscript, the study will be ready for publication.

Major Comments: 1) "First one, which the authors themselves have briefly mentioned, the climate and sea level rise projections used here are relatively old-dated! Publishing research based on 4th IPCC assessment report, while IPCC 5th has been around for years and IPCC 6th is coming out soon, needs a valid justification. To me, saying "was the only climate product, at the start of this project" is not enough. I expect a good justification that the current results are still useful for the audience. "

Response: We agree with the reviewer that using a more recent climate scenario could increase the strength of manuscript. Unfortunately, this project has required significant time and was started before/early in the IPCC 5th assessments availability. The choice of NARCAAP, and therefore the IPCC 4th assessment's climate scenarios, was based on data availability at the time of the studies start. We note in the manuscript:

"There are other downscaled climate products available (e.g., MACA; Abatzoglou [2013]) that are based on more current IPCC 5th assessment scenarios; however, NARCCAP was the only climate product, at the start of this project, that provided the necessary offshore coverage with the higher resolution RCM. Most other products are masked so that data are only available on land surfaces while this project required information across the ocean as well."

Additionally, the climate scenario used in the paper is consistent with current climate
trends and is well within the variability of IPCC 5th assessment scenarios. Therefore, we would argue that the conclusions from the study are still valid and a useful contribution to the scientific community. To clarify this with the reader we have included the following paragraph in the revised manuscript,

"While the usage of NARCCAP data forces this project's reliance an older climate scenario, this does not mean that results are out of alignment with current climate projections. Rather, the A2 SRES scenario is well within the variability of the new scenarios framework of the IPCC 5th assessment. A direct comparison of 4th and 5th IPCC assessment climate scenarios is impossible due to a conceptual change in how scenarios are handled [Nakicenovic et al., 2014; O'Neil et al., 2013]. However, work by Van Vuuren and Carter [2013] has shown that the A2 SRES scenario approximately maps to the representative concentration pathway (RCP) 8,5 and shared socio-economic pathway (SSP) 3 scenario. Since the publication of the IPCC 4th assessment, baseline emissions have been within the range presented within the SRES scenarios [IPCC, 2007] with emissions tracking closer to the higher range of scenarios [The Copenhagen Diagnosis, 2009]. This supports the usage of the A2 scenario for near term projections."

Utilized SLR projections were based on a study expanding on IPCC 4th assessment projections [NRC,2012]. While it would have been possible to utilize IPCC 5th assessment SLR projections, the NRC report provides local SLR estimates for the Pacific Coast. Vertical land motion in particular is very important at the study sites and it was decided that using slightly older local projections would be of a higher accuracy than a newer global projection. Additionally, using IPCC 4th assessment SLR projections provided consistency rather than mixing and matching forcing across assessments.

2) "At the end of section 4 (Lines 1-9 in Page 9), the non-linear interactions between SLR and tide/surge (i.e. Wahl, 2017, Sea-level rise and storm surges, relationship status: complicated!, Environ. Res. Lett., https://doi.org/10.1088/1748-9326/aa8eba ; Devlin et al., 2017, Coupling of sea level and tidal range changes, with implications

for future water levels, Scientific Reports, vol 7, 17021) should be highlighted that are missed here.

Response: We thank the reviewer for pointing out that this section is somewhat misleading in saying that non-stationarity is removed through the use of a variable datum. Rather this is an assumption that is discussed further in section 6.4.3. We have included an additional explanation of nonstationarity as a function of non-linear interactions within section 6.4.3. as well as acknowledged this assumption in the section discussed by the reviewer.

3) "Also, other alternatives for non-stationary frequency analysis should be explained as well for interested readers (i.e. Cheng L., AghaKouchak A., Gilleland E., Katz R.W., 2014, Non-stationary Extreme Value Analysis in a Changing Climate, Climatic Change, 127(2), 353-369, doi: 10.1007/s10584-014-1254-5.).

Response: We thank the reviewer for pointing out that we have not directly discussed a non-stationary approach. We have modified the paragraph starting at Page 8, L 37 to read as follows:

"Nonstationary extreme value analysis has recently seen a wide range of applications to coastal problems [Corbella and Stretch, 2012; Katz, 2013; Wahl and Chambers, 2016; Wahl et al., 2015]. Nonstationarity is generally incorporated within the GEV (Generalized Extreme Value) model by using time dependent parameters either as a linear or exponential function [Cheng et al., 2014; Ruggiero et al., 2010], a cyclical trigonometric function [Mendez et al., 2008; Minguez et al., 2010] or as a more complicated function of covariates [Weisse et al., 2014]. This study chooses an alternative to a nonstationary GEV approach primarily due to the format of the data from this study. WL data are output from the model in reference to MSL. This results in a WL time series that does not show any discontinuity or trend from changing sea level, a signal that would only be visible if viewing WLs relative to a non-tidal datum. This approximate stationarity, as a function of datum, makes it possible to separate the calculation of RIs

and the nonstationarity of the time series. This avoids the complications of fitting a non-stationary GEV (Generalized Extreme Value distribution) and the corresponding loss of degrees of freedom from estimating the nonstationary trend from the data. Furthermore, most nonstationary GEV analyses are forced to use a priori simplistic functions due to limited degrees of freedom. This approach allows a more complicated trend that follows experienced SLR (approximately cubic for this study). This approach of treating the resulting MSL timeseries as stationary is an assumption/simplification and is discussed further in the section 6.4.3."

4) "Page 4, L9-10: the hindcast and forecast periods are not the same length. This makes them incomparable, right¿'

Response: We acknowledge that ideally the hindcast and forecast periods would be the same length for statistical comparability between the two segments. Unfortunately, this was impossible due to data availability and the requirements for the various modeling components. We would argue that the two periods are still comparable, although with different uncertainty in extreme value estimates for the two periods. This is well exhibited in Figure 5 through the difference in confidence interval width between the historic and future period RI curves. We agree with the reviewer that this mismatch in confidence between estimates is unfortunate, but it was a compromise that we had to make in incorporating so many modeling components with differing data needs.

Minor Comments:

1) "Page 1, L 31-32: There are some studies to be cited here, i.e. Jay et al., 2016, Tidal-Fluvial and Estuarine Processes in the Lower Columbia River: II. Water Level Models, Floodplain Wetland Inundation, and System Zones, Estuaries and Coasts, Volume 39, Issue 5, pp 1299–1324; and the references therein."

Response: We thank the reviewer for bringing this article to our attention. We have added the reference to the revised manuscript.

2) "Page 2, L18: a relevant recent citation Gallien, et al., 2018, Coastal Flood Modeling Challenges in Defended Urban Backshores, Geosciences 2018, 8(12), 450; https://doi.org/10.3390/geosciences8120450."

Response: We thank the reviewer for pointing us to this excellent article and we have added the reference to the revised manuscript.

3) Page 6, L4: Please provide more details about the quantile mapping technique used here.

Response: While we agree with the reviewer that a clear identification of the utilized bias correction methods is important for transparency in the study's methodology, we think that a full description is beyond the scope of the paper. As an alternative we note that the utilized procedures are well described in a previous paper, Parker and Hill, 2017. We have added the following to the revised manuscript to direct interested readers to this resource:

"A full description of the utilized bias correction procedure, both the bivariate method utilized for wave modeling and the univariate method used for other variables, is beyond the scope of this paper. Instead, the reader is directed to Parker and Hill., [2017] for a more detailed description."

We hope that the reviewer finds this as an acceptable alternative to a significant lengthening of the article text length to descript the bias correction methodology.

We once again would like to the thank the reviewer for their time and effort. We think that the resulting paper is significantly stronger than the original due to the careful input of the reviewer. Thank you.

Please also note the supplement to this comment: https://www.nat-hazards-earth-syst-sci-discuss.net/nhess-2018-383/nhess-2018-383-AC1-supplement.pdf

---

## Referee Comment (RC2) · Bruno Merz (Referee) · 11 May 2019

This is a highly interesting study, using coupled, high-resolution and long-term modeling for assessing the flood hazard to 2 estuaries. The study is well motivated as it argues that the nonlinear interactions between the different drivers of flooding are not well understood in estuaries (in general) and at the US Pacific NW coast (in particular). I feel that the main novelty is the process-based modeling framework for analyzing climate change impacts to the different forcings of estuarine flooding. Some interesting conclusions are drawn, which challenge widespread assumptions, such as the bathtub approximation and adding an uncoupled high tide and high non-tidal residual to obtain

(compound) flooding magnitudes.

MAJOR COMMENTS:

- Section 3.4: Monthly Mean Sea-Level Anomalies (MMSLA) are modeled via a regression approach. Although I do not critize the regression approach, I wonder whether the variability of the modeled MMSLA agrees with the abserved one. Regression provides smoother responses, but the variability might be important when one looks at extreme water levels. Hence, is this regression step a source for underestimation of extremes? This should be clarified.

- Effective RIs and assumptions about nonstationarity: I would like to see a more accessible presentation of the concept of the study in relation to nonstationarity. I have not understood the concept. For example: * Section 4: Here it is explained that it is possible to separate the calculation of RIs and the nonstationarity of the time series. I do not fully understand what this means: Do you assume that the nonstationary is only a consequence of SLR? But other changes related to climate, such a changes in the wave climate, could also introduce changes in extreme water levels, right? I am confused and would like to see a clearer explanation. * Page 8, Line 33: I do not understand what it means that one can "... add the amount of nonstationarity (for this study, SLR) corresponding to the year of interest...". I guess this remark is related to the privious one. * Section 5.1, last paragraph and Figure 6: Again I do not understand the explanation of effective RIs and the locations of intersection between historic and future effective RIs. Why are the historic water levels higher than the future water levels when we have SLR?

- Future period: I am (partially) confused about the definition of 'future'. Please be clear about the future period (but also about the historic period) in the abstract, text and figures. For example, it would be good to give the 2 periods (historic: 1979-1999; future: 2041-2070) already in the abstract. Further, Fig. 9 says in the legend that the Flood Zone in 2100 is shown (although simulations have not been performed for this

period!) and the effective RI WL for the year 2050.

- One of the main conclusions is that extreme water levels are generally compound events, i.e. (in my understanding) the joint occurrence of different drivers, where the coupling between processes should be taken into account. On the other hand, the comparison with the FEMA flood zones seems to demonstrate that a simpler approach (I assume that the FEMA flood zones are not based on such a sophisticated model setup) leads to very similar results. Doesn't this invalidate your conclusion about the importance of compound events and the necessity to use coupled models?

- One of the limitations, acknowledged by the authors, is that the uncertainty is not included. They argue that one cannot use ensembles for this kind of complex model setup. I can follow their argument, but I would like to see a frank discussion about this tradeoff: Given limited resources, should I go for simpler models & ensembles or complex models without uncertainty quantification? I understand that there is no general answer to this question, but maybe the authors can discuss the different arguments and "recommend" what one could or should do (maybe considering different purposes, e.g. planning of flood defense).

MINOR COMMENTS:

- Page 2, Line 1 and Page 12, Line 19: Please use only literature which has been published.

- Page 4, Line 25: I am surprised to learn that the RCMs have a super high resolution: "... Spatial resolution for models within NARCCAP is 50 m for RCM variables...". I just want to make sure that this is not a typo.

---

## Author Comment (AC2) · 7 Jun 2019

For additional readability of responses, please consult the supplemental Pdf. For this response, each reviewer comment is listed in quotes and followed by our response. We sincerely thank the reviewer for their time and believe that the resulting manuscript has been greatly improved through the review process. Thank you!

Summary: "This is a highly interesting study, using coupled, high-resolution and long-term modeling for assessing the flood hazard to 2 estuaries. The study is well motivated as it argues that the nonlinear interactions between the different drivers of flooding are not well understood in estuaries (in general) and at the US Pacific NW coast (in

particular). I feel that the main novelty is the process-based modeling framework for analyzing climate change impacts to the different forcings of estuarine flooding. Some interesting conclusions are drawn, which challenge widespread assumptions, such as the bathtub approximation and adding an uncoupled high tide and high non-tidal residual to obtain (compound) flooding magnitudes."

Response: We thank the reviewer for the assessment of the article. We appreciate the clearly significant effort that the reviewer has spent on the article and hope that, with the modifications detailed below, the article will be acceptable for publication.

Major Comments: 1) "Section 3.4: Monthly Mean Sea-Level Anomalies (MMSLA) are modeled via a regression approach. Although I do not criticize the regression approach, I wonder whether the variability of the modeled MMSLA agrees with the observed one. Regression provides smoother responses, but the variability might be important when one looks at extreme water levels. Hence, is this regression step a source for underestimation of extremes? This should be clarified."

Response: We agree with the reviewer in that the utilized regression approach adds an additional layer of uncertainty into estimates of WLs. This is especially true concerning extremes as the regression is unable to reproduce maximums in MMSLA's. This said, a computational approach (for example a 3-D regional scale coastal model like ROMS) would be extremely computationally expensive. This approach would likely still be imperfect when forced by coarse AOGCM outputs. The taken statistical approach allows a compromise of modelling the leading order effect of MMSLA's on WLs. While admittedly imperfect, this was required computationally and a methodological decision that is well represent in the literature.

While there is uncertainty in MMSLA's, most of the key conclusions from this paper are not affected by this issue. We are mainly interested in changes from historic to future conditions and the bias induced by using regression for MMSLA's (likely a bias low as pointed out by the reviewer) should be similar for both periods. In other words, our bulk

estimates of RIs might be biased but the difference from climate change should remain valid. A similar argument could be made about spatial variability in return intervals as MMSLA's effect RI's in an approximately spatially uniform manner (see figure 10).

We agree with the reviewer that the current version of the article does not properly discuss this important point and we have added text discussing the problem in section 3.4, 5.3, and 6.1.

2) "Effective RIs and assumptions about nonstationarity: I would like to see a more accessible presentation of the concept of the study in relation to nonstationarity. I have not understood the concept. For example: * Section 4: Here it is explained that it is possible to separate the calculation of RIs and the nonstationarity of the time series. I do not fully understand what this means: Do you assume that the nonstationary is only a consequence of SLR? But other changes related to climate, such a changes in the wave climate, could also introduce changes in extreme water levels, right? I am confused and would like to see a clearer explanation."

Response: The reviewer is correct that, in the approach taken by this paper for nonstationary RIs, there is an implicit assumption that removing SLR will make the timeseries "approximately" stationary. This is definitely an assumption and is discussed in detail in section 6.4.3. We agree with the reviewer that this is not clear in the original version of the manuscript and have added additional text explicitly detailing this assumption in section 4. We have additionally added a statement referring the reader to section 6.4.3 for additional discussion.

* "Page 8, Line 33: I do not understand what it means that one can "... add the amount of nonstationarity (for this study, SLR) corresponding to the year of interest...". I guess this remark is related to the previous one."

Response: We agree with the reviewer that this section is difficult to understand without a visual. In an earlier version of the paper, we had placed the effective return interval curve results here to help with explaining what is meant by this text. During interval

review, this was flagged as inappropriate since we were displaying results outside of the "results" section. We also thought about inserting a generic figure here for explanatory purposes, but this was also decided against as the article is already long in terms of figures and word count.

We have significantly revised this section with hopes of increasing overall clarity. We have removed the section quoted by the reviewer and attempted to replace it with a more understandable version. That said, if the reviewer thinks that a figure would still help, we could either move a results figure here or add a generic figure.

* "Section 5.1, last paragraph and Figure 6: Again I do not understand the explanation of effective RIs and the locations of intersection between historic and future effective RIs. Why are the historic water levels higher than the future water levels when we have SLR?"

Response: We agree with the reviewer that this presentation of results is somewhat non-intuitive. In considering this figure we were worried that, as detailed in the comment, the immediate take away would be that future flooding is less than historic. This is definitely not true. The future RI curve represents recurrence water level events without SLR included. We decided to present the data in this way for two reasons. The first is because we feel that it shows a valuable outcome of the study, specifically ("future RIs as a function of only changing forcing (no SLR)"). The second is that to plot non-stationary Future recurrence intervals (SLR included) on the same plot as stationary historic recurrence intervals is conceptually problematic. Stationary and nonstationary recurrence intervals are not directly equivalent so we feel this would be incorrect. Instead we plot the stationary version of the Future recurrence intervals (that with SLR not included) to emphasize this point.

To clarify this further, we have added (No SLR) to the Future symbol in the legend of Figure 6. We have additionally augmented the text in the description of the figure to further emphasize this point (as well as our reasons for presenting the data in this

way). This said, if the reviewer has any further suggestions for how we could improve clarity, we would very much like to incorporate it in the paper. This is an important detail that we don't want to get lost. 3) "Future period: I am (partially) confused about the definition of 'future'. Please be clear about the future period (but also about the historic period) in the abstract, text and figures. For example, it would be good to give the 2 periods (historic: 1979-1999; future: 2041-2070) already in the abstract. Further, Fig. 9 says in the legend that the Flood Zone in 2100 is shown (although simulations have not been performed for this period!) and the effective RI WL for the year 2050."

Response: The reviewer here brings up a point of confusion that, in retrospect, we have not explained sufficiently in the paper. Simulations are for the 2 periods mentioned (historic: 1979-1999: future: 2041-2070). From these it is possible to calculate return intervals events with an associated recurrence interval (in this study ranging from 2 – 100 year). Unfortunately, in terms of clarity, our use of effective return intervals adds an additional dimension of time to the concept of return intervals. We have significantly re-worked the section describing effective return intervals with hopes of improving clarity. We have taken the reviewers suggestion to add the 2 periods of simulation to the abstract. We have additionally changed the legend in Figure 9 to say 100-yr Flood Zone (Hist.) and 100-yr Flood zone (2100). We have also added text hopefully clarifying this point. As to the reviewer's comment regarding the plotting of the flood zone in 2100, this is a calculable quantity even though we don't have simulations for this period. A similar argument could be made for the common extreme value analysis (EVA) procedure of calculating a 100-year event without 100 years of simulations. Just as in for that case (EVA), we are using a statistical model. This allows us to make a prediction for the 100 year return interval flood zone in 2100, of course under the constraints of that model (which is just a model and a simplification). The key assumption is (as pointed out in the reviewer's other comments) SLR is the source of nonstationarity. We have tried to emphasize this more in the revised text of the article as well as included an in depth discussion of this assumption (section 6.4.3).

4) "One of the main conclusions is that extreme water levels are generally compound events, i.e. (in my understanding) the joint occurrence of different drivers, where the coupling between processes should be taken into account. On the other hand, the comparison with the FEMA flood zones seems to demonstrate that a simpler approach (I assume that the FEMA flood zones are not based on such a sophisticated model setup) leads to very similar results. Doesn't this invalidate your conclusion about the importance of compound events and the necessity to use coupled models?"

Response: Similarly to the reviewer, we were surprised to find out that the calculated 100-year flood zone for this study and FEMAs were so similar. Unfortunately, the report detailing FEMA's approach is quite opaque, so it is difficult to determine the exact differences between the two methodologies. The FEMA assessment for Coos Bay was performed fairly recently (2014) so does appear to include some process coupling and more modern approaches for estuary systems. This said we cannot conclusively say to what extent.

As an additional point of consideration, this figure only allows for a planform comparison in flood areas. Differences in flood areas are strongly controlled by gradients in terrain. For example, a 1 meter difference in flood elevation can result in a significant difference in flood area for flat regions. Coos Bay, on the other hand, is hemmed in by mountains and so large differences in flood elevations may not result in a significant change in flood area (especially at the scale of the map in the discussed figure).

We decided to include this figure in the paper as we found it to be a valuable validation tool. We felt that validation was a weakness of the study and that this figure provided some confidence moving forward. While we agree with the reviewer's discussion, we still find the figure to be valuable in this regard.

5) "One of the limitations, acknowledged by the authors, is that the uncertainty is not included. They argue that one cannot use ensembles for this kind of complex model setup. I can follow their argument, but I would like to see a frank discussion about this

tradeoff: Given limited resources, should I go for simpler models & ensembles or complex models without uncertainty quantification? I understand that there is no general answer to this question, but maybe the authors can discuss the different arguments and "recommend" what one could or should do (maybe considering different purposes, e.g. planning of flood defense)."

Response: This is a very pertinent comment by the reviewer and something that, through the process of this study, we spent a lot of time thinking about. We did not include this discussion in the original manuscript as we were unsure if it was too divergent from the purpose of this paper. This said, we have happily included a brief discussion section in the revised manuscript (section 6.4.2).

Minor Comments: 1) "Page 2, Line 1 and Page 12, Line 19: Please use only literature which has been published."

Response: This reference has been removed in the revised manuscript.

2) "Page 4, Line 25: I am surprised to learn that the RCMs have a super high resolution: "... Spatial resolution for models within NARCCAP is 50 m for RCM variables...". I just want to make sure that this is not a typo."

Response: As the reviewer correctly assumed, this was a typo. The RCM resolution in NARCCAP is 50 km. This has been corrected in the revised manuscript.

Please also note the supplement to this comment:
https://www.nat-hazards-earth-syst-sci-discuss.net/nhess-2018-383/nhess-2018-383-AC2-supplement.pdf

―――――――――――――――――

---

## Author Response (AR1)

Dear Editor, Associate Editor, and Reviewers.

5  Thank you for your assessment of nhess_2018_383, "The Effects of Changing Climate on Estuarine Water Levels: A United States Pacific Northwest Case Study" by Parker, Hill, García-Medina, and Beamer. We sincerely appreciated the obvious effort that the reviewers have put into the paper and their resulting suggested areas for improvement. As detailed in the open discussion of the manuscript, we have incorporated all reviewer comments into a second draft which we now submit for your consideration. In the following section we describe
10  our improvements to the manuscript as guided by reviewer comments. These comments are similar to those from the open discussion although with some changes. This is followed by the revised manuscript.

Overall, we were in agreement with most reviewer's comments and found their incorporation into the revised manuscript relatively minor. The majority of comments regarded improving clarity and we have tried to re-write
15  the mentioned sections to the best of our abilities. This said, we welcome any additional comments by the reviewers to further improve clarity.

We hope that you will find this revised manuscript suitable for publication in Natural Hazards and Earth System Sciences and we look forward to hearing from you in the near future.
20
Kind Regards,

25  *Kai Parker*

Kai Parker (Ph.D.)
School of Civil and Construction Engineering - Oregon State University

30  Contact Information
**Address:** 1717 7th St.
                Los Osos, CA 93402. USA
**Phone:**    +1 (775) 560-3210
**Email:**    parkerk@oregonstate.edu

**Anonymous Referee #1**

**Recommendation:** Major revision

**Summary:**

5   *"The authors have proposed a modeling framework to quantify the extreme estuarine water levels (WL) under climate change. Their framework, taking oceanic, atmospheric and hydrologic processes into account estimates extreme water levels in two estuaries of US Pacific Northwest. The idea of integrated modeling that couples processes across scales to estimate extreme estuarine water levels is interesting, and I believe this idea deserves publishing in NHESS, however after a major revision. There are a few*
10  *issues with this version that I explain below:* "

Response: We appreciate the reviewer's assessment of the paper. We hope that, with the reviewer's comments integrated into the new manuscript, the study will be ready for publication.

**Major Comments:**

15  1)   *"First one, which the authors themselves have briefly mentioned, the climate and sea level rise projections used here are relatively old-dated! Publishing research based on 4th IPCC assessment report, while IPCC 5th has been around for years and IPCC 6th is coming out soon, needs a valid justification. To me, saying "was the only climate product, at the start of this project" is not enough. I expect a good justification that the current results are still useful for the audience. "*

20  Response: We agree with the reviewer that using a more recent climate scenario could increase the strength of manuscript. Unfortunately, this project has required significant time and was started before the publication of the IPCC 5th assessment. This said, the climate scenario used in the paper is consistent with current climate trends and is well within the variability of IPCC 5th assessment scenarios. Therefore, we would argue that the conclusions from the study are still valid and a useful
25  contribution to the scientific community. To clarify this with the reader we have included the following paragraph in the revised manuscript,

    "While the usage of NARCCAP data forces this project's reliance an older climate scenario, this does not mean that results are out of alignment with current climate projections. Rather, the A2
30  SRES scenario is well within the variability of the new scenarios framework of the IPCC 5th assessment. A direct comparison of 4th and 5th IPCC assessment climate scenarios is impossible due to a conceptual change in how scenarios are handled [Nakicenovic et al., 2014; O'Neill et al., 2013]. However, work by Van Vuuren and Carter [2014] has shown that the A2 SRES scenario

approximately maps to the representative concentration pathway (RCP) 8,5 and shared socio-economic pathway (SSP) 3 scenario. Since the publication of the IPCC 4th assessment, baseline emissions have been within the range presented within the SRES scenarios [IPCC, 2007] with emissions tracking closer to the higher range of scenarios [The Copenhagen Diagnosis, 2009]. This supports the usage of the A2 scenario for near term projections."

Utilized SLR projections were based on a study expanding on IPCC 4th assessment projections [NRC,2012]. While it would have been possible to utilize IPCC 5th assessment SLR projections, the NRC report provides local SLR estimates for the Pacific Coast. Vertical land motion in particular is very important at the study sites and it was decided that using slightly older local projections would be of a higher accuracy than a newer global projection.

2) *"At the end of section 4 (Lines 1-9 in Page 9), the non-linear interactions between SLR and tide/surge (i.e. Wahl, 2017, Sea-level rise and storm surges, relationship status: complicated!, Environ. Res. Lett., https://doi.org/10.1088/1748-9326/aa8eba ; Devlin et al., 2017, Coupling of sea level and tidal range changes, with implications for future water levels, Scientific Reports, vol 7, 17021) should be highlighted that are missed here.*

Response: We thank the reviewer for pointing out that this section is somewhat misleading in saying that non-stationarity is removed through the use of a variable datum. Rather this is an assumption that is discussed further in section 6.4.3. We have included an additional explanation of nonstationarity as a function of non-linear interactions within section 6.4.3. as well as acknowledged this assumption in the section discussed by the reviewer.

3) *"Also, other alternatives for non-stationary frequency analysis should be explained as well for interested readers (i.e. Cheng L., AghaKouchak A., Gilleland E., Katz R.W., 2014, Non-stationary Extreme Value Analysis in a Changing Climate, Climatic Change, 127(2), 353-369, doi: 10.1007/s10584-014-1254-5.)."*

Response: We thank the reviewer for pointing out that we have not directly discussed a non-stationary approach. We have modified the paragraph starting at Page 8, L 37 to read as follows:

"Nonstationary extreme value analysis has recently seen a wide range of applications to coastal problems [Corbella and Stretch, 2012; Katz, 2013; Wahl and Chambers, 2016; Wahl et al., 2015]. Nonstationarity is generally incorporated within the statistical model by using time dependent parameters either as a linear or exponential function [Cheng et al., 2014; Ruggiero et al., 2010], a cyclical trigonometric function [Méndez et al., 2008; Mínguez et al., 2010] or as a more complicated function of covariates [Méndez et al., 2007; Weisse et al., 2014]. Even for stationary extreme value analysis there is a range of commonly used statistical models with a corresponding uncertainty as a result of the chosen methodology [Wahl et al., 2017]. This study uses a stationary Generalized Extreme Value (GEV) approach primarily due to the format of the data from this study. WL data are output from the model in reference to MSL. This results in a WL time series

that does not show any discontinuity or trend from changing sea level, a signal that would only be visible if viewing WLs relative to a non-tidal datum. This approximate stationarity, as a function of datum, makes it possible to separate the calculation of RIs and the nonstationarity of the time series. This avoids the complications of fitting a nonstationary GEV and the corresponding loss of degrees of freedom from estimating the nonstationary trend from the data. Furthermore, most nonstationary GEV analyses are forced to use a priori simplistic functions due to limited degrees of freedom. This approach allows a more complicated trend that follows experienced SLR (approximately cubic for this study). This approach of treating the resulting MSL timeseries as stationary is an assumption/simplification and is discussed further in the section 6.4.3."

4) *"Page 4, L9-10: the hindcast and forecast periods are not the same length. This makes them incomparable, right?"*

Response: We acknowledge that ideally the hindcast and forecast periods would be the same length for statistical comparability between the two segments. Unfortunately, this was impossible due to data availability and the requirements for the various modeling components. We would argue that the two periods are still comparable, although with different uncertainty in extreme value estimates for the two periods. This is well exhibited in Figure 5 through the difference in confidence interval width between the historic and future period RI curves. We agree with the reviewer that this mismatch in confidence between estimates is unfortunate, but it was a compromise that we had to make in incorporating so many modeling components with differing data needs.

**Minor Comments:**

1) *"Page 1, L 31-32: There are some studies to be cited here, i.e. Jay et al., 2016, Tidal-Fluvial and Estuarine Processes in the Lower Columbia River: II. Water Level Models, Floodplain Wetland Inundation, and System Zones, Estuaries and Coasts, Volume 39, Issue 5, pp 1299–1324; and the references therein."*

Response: We thank the reviewer for bringing this article to our attention. We have added the reference to the revised manuscript.

2) *"Page 2, L18: a relevant recent citation Gallien, et al., 2018, Coastal Flood Modeling Challenges in Defended Urban Backshores, Geosciences 2018, 8(12), 450; https://doi.org/10.3390/geosciences8120450."*

Response: We thank the reviewer for pointing us to this excellent article and we have added the reference to the revised manuscript.

3) Page 6, L4: Please provide more details about the quantile mapping technique used here.

Response: While we agree with the reviewer that a clear identification of the utilized bias correction methods is important for transparency in the study's methodology, we think that a full description is beyond the scope of the paper. As an alternative we note that the utilized procedures are well described in a previous paper, Parker and Hill, 2017. We have added the following to the revised manuscript to direct interested readers to this resource:

"A full description of the utilized bias correction procedure, both the bivariate method utilized for wave modeling and the univariate method used for other variables, is beyond the scope of this paper. Instead, the reader is directed to Parker and Hill., [2017] for a more detailed description."

We hope that the reviewer finds this as an acceptable alternative to a significant lengthening of the article text length to descript the bias correction methodology.

**Bruno Merz (Referee #2)**

**Recommendation:** Major revision

**Summary:**

5 *"This is a highly interesting study, using coupled, high-resolution and long-term modeling for assessing the flood hazard to 2 estuaries. The study is well motivated as it argues that the nonlinear interactions between the different drivers of flooding are not well understood in estuaries (in general) and at the US Pacific NW coast (in particular). I feel that the main novelty is the process-based modeling framework for analyzing climate change impacts to the different forcings of estuarine flooding. Some interesting*
10 *conclusions are drawn, which challenge widespread assumptions, such as the bathtub approximation and adding an uncoupled high tide and high non-tidal residual to obtain (compound) flooding magnitudes."*

Response: We thank the reviewer for the assessment of the article. We appreciate the clearly significant effort that the reviewer has spent on the article and hope that, with the modifications detailed below, the
15 article will be acceptable for publication.

**Major Comments:**

1) *"Section 3.4: Monthly Mean Sea-Level Anomalies (MMSLA) are modeled via a regression approach. Although I do not criticize the regression approach, I wonder whether the variability*
20 *of the modeled MMSLA agrees with the observed one. Regression provides smoother responses, but the variability might be important when one looks at extreme water levels. Hence, is this regression step a source for underestimation of extremes? This should be clarified."*

Response: We agree with the reviewer in that the utilized regression approach adds an additional layer of uncertainty into estimates of WLs. This is especially true concerning extremes as the regression is
25 unable to reproduce maximums in MMSLA's. This said, the decision to use regression was primarily motivated by available data from the paired global/regional climate model. A modeling approach that could more accurately reproduce MMSLA's (for example a 3-D regional scale coastal model like ROMS) would be: a) computationally infeasible b) not necessarily any better when forced by coarse AOGCM outputs and c) potentially impossible depending on available water column boundary
30 conditions for the AOGCM implementations.
While there is high uncertainty in used MMSLA's, most of the key conclusions from this paper are not affected by this issue. We are mainly interested in changes from historic to future conditions and the

bias induced by using regression for MMSLA's (likely a bias low as pointed out by the reviewer) should be the same for both periods. In other words, our bulk estimates of RIs might be biased but the difference from climate change should remain valid. A similar argument could be made about spatial variability in return intervals as MMSLA's effect RI's in an approximately spatially uniform manner (see figure 10).

We agree with the reviewer that the current version of the article does not properly discuss this important point and we have added text discussing the problem in section 3.4, 5.3, and 6.1.

2) *"Effective RIs and assumptions about nonstationarity: I would like to see a more accessible presentation of the concept of the study in relation to nonstationarity. I have not understood the concept. For example:*

*Section 4: Here it is explained that it is possible to separate the calculation of RIs and the nonstationarity of the time series. I do not fully understand what this means: Do you assume that the nonstationary is only a consequence of SLR? But other changes related to climate, such a changes in the wave climate, could also introduce changes in extreme water levels, right? I am confused and would like to see a clearer explanation.*

Response: The reviewer is correct that, in the approach taken by this paper for nonstationary RIs, there is an implicit assumption that removing SLR will make the timeseries "approximately" stationary. This is definitely an assumption and is discussed in detail in section 6.4.3. We agree with the reviewer that this is not clear in the original version of the manuscript and have added additional text explicitly detailing this assumption in section 4. We have additionally added a statement referring the reader to section 6.4.3 for additional discussion.

*Page 8, Line 33: I do not understand what it means that one can "... add the amount of nonstationarity (for this study, SLR) corresponding to the year of interest...". I guess this remark is related to the previous one.*

Response: We agree with the reviewer that this section is difficult to understand without a visual. In an earlier version of the paper, we had placed the effective return interval curve results here to help with explaining what is meant by this text. During interval review, this was flagged as inappropriate since we were displaying results outside of the "results" section. We also thought about inserting a generic figure here for explanatory purposes, but this was also decided against as the article is already long in terms of figures and word count.

We have significantly revised this section with hopes of increasing overall clarity. We have removed the section quoted by the reviewer and attempted to replace it with a more understandable version. That

said, if the reviewer thinks that a figure would still help, we could either move a results figure here or add a generic figure.

*\* Section 5.1, last paragraph and Figure 6: Again I do not understand the explanation of effective RIs and the locations of intersection between historic and future effective RIs. Why are the historic water levels higher than the future water levels when we have SLR?"*

Response: We agree with the reviewer that this presentation of results is somewhat non-intuitive. In considering this figure we were worried that, as detailed in the comment, the immediate take away would be that future flooding is less than historic. This is definitely not true. The future RI curve represents recurrence water level events without SLR included. We decided to present the data in this way for two reasons. The first is because we feel that it shows a valuable outcome of the study, specifically ("future RIs as a function of only changing forcing (no SLR)"). The second is that to plot nonstationary Future recurrence intervals (SLR included) on the same plot as stationary historic recurrence intervals is conceptually problematic. Stationary and nonstationary recurrence intervals are not directly equatable so we feel this would be incorrect. Instead we plot the stationary version of the Future recurrence intervals (that with SLR not included).

To clarify this further, we have added (No SLR) to the Future symbol in the legend of Figure 6. We have additionally augmented the text in the description of the figure to further emphasize this point (as well as our reasons for presenting the data in this way). This said, if the reviewer has any further suggestions for how we could better get this point across better, we would very much like to incorporate it in the paper. This is an important detail that we don't want to get lost.

3) *"Future period: I am (partially) confused about the definition of 'future'. Please be clear about the future period (but also about the historic period) in the abstract, text and figures. For example, it would be good to give the 2 periods (historic: 1979-1999; future: 2041-2070) already in the abstract. Further, Fig. 9 says in the legend that the Flood Zone in 2100 is shown (although simulations have not been performed for this period!) and the effective RI WL for the year 2050."*

Response: The reviewer here brings up a point of confusion that, in retrospect, we have not explained sufficiently in the paper. Simulations are for the 2 periods mentioned (historic: 1979-1999: future: 2041-2070). From these it is possible to calculate return intervals events with an associated recurrence interval (in this study ranging from 2 – 100 year). Unfortunately, in terms of clarity, our use of effective return intervals adds an additional dimension of time to the concept of return intervals. We have significantly re-worked the section describing effective return intervals with hopes of improving clarity. We have taken the reviewers suggestion to add the 2 periods of simulation to the abstract. We

have additionally changed the legend in Figure 9 to say 100-yr Flood Zone (Hist.) and 100-yr Flood zone (2100). We have also added text hopefully clarifying this point.

As to the reviewers comment regarding the plotting of the flood zone in 2100, this is a calculatable quantity even though we don't have simulations for this period. A similar argument could be made for the common extreme value analysis (EVA) procedure of calculating a 100-year event without 100 years of simulations. Just as in for that case (EVA), we are using a statistical model. This allows us to make a prediction for the 100 year return interval flood zone in 2100, of course under the constraints of that model (which is just a model and a simplification). The key assumption is (as pointed out in the reviewers other comments) SLR is the source of nonstationarity. We have tried to emphasize this more in the revised text of the article as well as included an in depth discussion of this assumption (section 6.4.3).

4) *"One of the main conclusions is that extreme water levels are generally compound events, i.e. (in my understanding) the joint occurrence of different drivers, where the coupling between processes should be taken into account. On the other hand, the comparison with the FEMA flood zones seems to demonstrate that a simpler approach (I assume that the FEMA flood zones are not based on such a sophisticated model setup) leads to very similar results. Doesn't this invalidate your conclusion about the importance of compound events and the necessity to use coupled models?"*

Response: In further consideration prompted by the reviewer, we have decided to remove this figure (comparing FEMA flood zones to our results). The figure was originally intended as a further validation of the modeling framework. It doesn't accomplish this particularly well as it is a comparison of planform flood areas at a zoomed out spatial scaled. Coos Bay is mountainous with steep terrain near the shore so even large differences in predicted extreme water levels would not manifest as a significant difference in flooding area. We have decided to discuss this in the text and removed the figure from the manuscript.

5) *"One of the limitations, acknowledged by the authors, is that the uncertainty is not included. They argue that one cannot use ensembles for this kind of complex model setup. I can follow their argument, but I would like to see a frank discussion about this tradeoff: Given limited resources, should I go for simpler models & ensembles or complex models without uncertainty quantification? I understand that there is no general answer to this question, but maybe the authors can discuss the different arguments and "recommend" what one could or should do (maybe considering different purposes, e.g. planning of flood defense)."*

Response: This is a very pertinent comment by the reviewer and something that, through the process of this study, we spent a lot of time thinking about. We did not include this discussion in the original

manuscript as we were unsure if it was too divergent from the purpose of this paper. This said, we have happily included a brief discussion section in the revised manuscript (section 6.4.2).

**Minor Comments:**

5    1)  *"Page 2, Line 1 and Page 12, Line 19: Please use only literature which has been published."*

Response: This reference has been removed in the revised manuscript.

   2)  *"Page 4, Line 25: I am surprised to learn that the RCMs have a super high resolution: "... Spatial resolution for models within NARCCAP is 50 m for RCM variables...". I just want to make sure that this is not a typo."*

Response: As the reviewer correctly assumed, this was a typo. The RCM resolution in NARCCAP is 50 km. This has been corrected in the revised manuscript.

[revised manuscript text omitted]

---

## Author Response (AR2)

**Referee #1 – Hamed Moftakhari**

**Received:** 03 July 2019

**Recommendation:** Accept subject to minor revisions

**Summary:**

"Authors have done a great job in this revision. The manuscript should be ready for publication after some minor reference corrections:"

Response: We thank the reviewer for the kind words in terms of the revision. We hope that with the corrected references, this paper is ready for publication.

**Minor Comments:**

1)  "Page 2, Line 32: It should be "Moftakhari et al., 2015; 2017", while the appropriate citation is already provided in the reference list."

Response: We agree with the reviewer that the Maftakhari et al., 2015,2017 articles relating to "nuisance flooding" belong together. This said, we feel that these references are better placed at page 2, line 32-33. Judging from the reviewer's 2nd comment, we feel that the reviewer would most likely be in agreement.

2)  "Page 2, Line 23: In the sentence about "Compound Events" the authors meant to cite the following publication that is missed in the reference list:

> • Moftakhari, H. R., G. Salvadori, A. AghaKouchak, B. F. Sanders, and R. A. Matthew (2017), Compounding Effects of Sea Level Rise and Fluvial Flooding, Proceedings of the National Academy of Sciences, 114 (37), 9785-9790, DOI: 10.1073/pnas.1620325114.

The following citation would be appropriate here as well:
> • Moftakhari, H. R., Schubert, J., AghaKouchak, A., Matthew, R. A., & Sanders, B. F. (2019). Linking Statistical and Hydrodynamic Modeling for Compound Flood Hazard Assessment in Tidal Channels and Estuaries. Advances in Water Resources, https://doi.org/10.1016/j.advwatres.2019.04.009."

Response: We thank the reviewer for pointing out this error. As the reviewer correctly deduced, we intend to cite the mentioned Moftakhari et al., 2017 (*Compounding Effects of Sea Level Rise and Fluvial Flooding*) at this point and the Moftakhari et al., 2017 article (*Cumulative hazard: The case of nuisance flooding*) later at page 2, line 32-33. We have revised the manuscript with these corrected references (as well as added the missed reference to the reference list).

We also would like to thank the reviewer for pointing us to the additional article and have added this relevant reference to the paper. This is also a very exciting research direction that we have

similarly been pursuing. An additional thanks to the reviewer for showing us this recently released work!